# Generative Neural Operators through Diffusion Last Layer

**Sungwon Park** [1 2]  **Anthony Zhou** [2]  **Hongjoong Kim** [1]  **Amir Barati Farimani** [2]

## Abstract

Neural operators provide a powerful framework for learning discretization invariant mappings between function spaces, but standard deterministic models do not capture predictive uncertainty. We introduce *diffusion last layer* (DLL), a modular probabilistic output head for neural operator backbones. DLL represents target fields through an input dependent low rank expansion inspired by the Karhunen–Loève expansion and learns a conditional diffusion model over the corresponding coefficient space. This design enables efficient distributional modeling while preserving the structural advantages of operator learning. On stochastic PDE benchmarks with random forcing, DLL achieves strong distributional fidelity and performs competitively with pixel space and conventional latent diffusion baselines. In deterministic long horizon rollout tasks, DLL improves rollout stability over the underlying backbone and provides useful estimates of predictive uncertainty under compounding autoregressive errors. These results suggest that diffusion modeling in learned coefficient spaces offers a practical route to uncertainty aware neural operators. Code is available at  github.com/sungwpark/dll-no.

## 1. Introduction

To approximate function-to-function mappings with neural surrogates, neural operators have emerged as a general paradigm and have been studied extensively across diverse architectures (Kovachki et al., 2023; Azizzadenesheli et al., 2024). Representative approaches include DeepONet, which parameterizes operators via a branch trunk decomposition (Lu et al., 2021), and the Fourier neural operator (FNO), which employs global spectral convolutions to en-

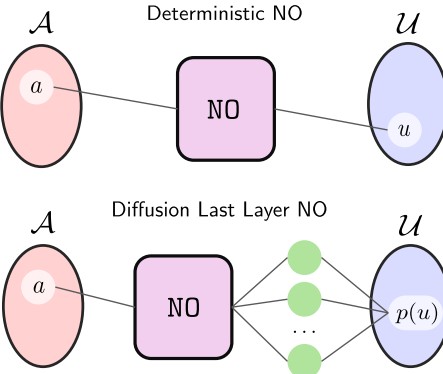

*Figure 1.* Deterministic vs. generative neural operators. A standard neural operator (top) maps an input field $a \in \mathcal{A}$ to a single prediction $u \in \mathcal{U}$. Our *Diffusion Last Layer* (bottom) turns the same backbone into a conditional generator by attaching a lightweight diffusion head, producing a full predictive distribution $p(u \,|\, a)$ over functions rather than a point estimate.

able discretization-robust learning (Li et al., 2020). Subsequent work has expanded this line of research through improved scalability (Tran et al., 2023), extensions to geometric and irregular domains (Li et al., 2023a;b), spherical and multiscale encoder–decoder operator designs (Bonev et al., 2023; Rahman et al., 2023), and attention-based operator models that better capture long-range dependencies in complex PDE systems (Li et al., 2023c; Hao et al., 2023; 2024; Herde et al., 2024).

In many scientific applications, the target field exhibits intrinsic randomness arising from stochastic forcing, unresolved subgrid scale effects, uncertain coefficients, or chaotic sensitivity, making accurate uncertainty quantification (UQ) essential for reliable surrogate modeling (Xiu, 2010). Motivated by this need, recent advances in scientific machine learning have developed probabilistic operator learning frameworks that model conditional distributions over functions, rather than returning deterministic point predictions (Psaros et al., 2023). Representative approaches include Bayesian neural operators (Lin et al., 2023; Weber et al., 2024; Magnani et al., 2025a;b), which aim to capture epistemic uncertainty via posterior inference over model parameters. Complementary lines of work instead construct generative operator surrogates that directly sample diverse

[1]Korea University, Seoul, South Korea [2]Carnegie Mellon University, Pittsburgh, USA. Correspondence to: Hongjoong Kim <hongjoong@korea.ac.kr>, Amir Barati Farimani <barati@cmu.edu>.

*Proceedings of the 43rd International Conference on Machine Learning*, Seoul, South Korea. PMLR 306, 2026. Copyright 2026 by the author(s).

realizations from the conditional solution distribution (Bülte et al., 2025), or train function space generative models using neural operator and related function space parameterizations (Rahman et al., 2022; Du et al., 2024; Hu et al., 2025; Shi et al., 2025).

Meanwhile, recent progress in diffusion models (Sohl-Dickstein et al., 2015; Ho et al., 2020; Song et al., 2021) and flow matching (Liu et al., 2023; Lipman et al., 2023; Albergo et al., 2025) for high-dimensional fields has provided a new route to probabilistic PDE surrogates. These generative frameworks have been actively explored in scientific forecasting and data assimilation, particularly under sparse or partial observations (Lippe et al., 2023; Shysheya et al., 2024; Huang et al., 2024). To improve scalability, latent generative formulations compress the solution state space while largely preserving predictive fidelity (Rozet et al., 2025; Zhou et al., 2025b). Related encoder based designs further extend these models to irregular domains and geometric settings, enabling conditional generation beyond regular grids (Zhou et al., 2025a; Wang et al., 2025).

In this work, we propose the diffusion last layer (DLL), a lightweight probabilistic module for neural operator backbones, as illustrated in Figure 1. DLL preserves the backbone's discretization invariance and geometry awareness while extending deterministic operators to conditional generative modeling. This is achieved by training the diffusion head in a low-dimensional coefficient space associated with input dependent basis functions produced by an operator encoder, rather than directly in pixel space. This enables efficient high resolution sampling while retaining the structural advantages of neural operators. Empirically, DLL captures stochastic PDE solution distributions under random forcing with strong distributional fidelity, and improves long horizon rollout stability with useful uncertainty estimates in deterministic settings.

## 2. Background

### 2.1. Operator Learning

Many scientific relationships, especially in physical simulation, can be formulated as *operators*:

$$\mathcal{G}^\dagger : \mathcal{A} \to \mathcal{U},$$

where $\mathcal{A}$ and $\mathcal{U}$ are function spaces and the output is determined by a deterministic rule (Kovachki et al., 2023; Azizzadenesheli et al., 2024).

For stochastic systems, it is natural to generalize deterministic operators to *stochastic operators*:

$$\mathcal{G}^\ddagger : \mathcal{A} \to \mathcal{P}(\mathcal{U}),$$

where $\mathcal{P}(\mathcal{U})$ denotes the space of probability measures on $\mathcal{U}$. Equivalently, $\mathcal{G}^\ddagger(a)$ specifies a conditional distribution

over outputs $u \mid a$, so that a single input function induces a distribution of possible solutions rather than a unique realization (Bülte et al., 2025).

Deterministic operators are recovered as a special case: if $\mathcal{G}^\ddagger(a) = \delta_{\mathcal{G}^\dagger(a)}$, then the conditional law collapses to a Dirac measure concentrated at $\mathcal{G}^\dagger(a)$.

We adopt a unified probabilistic formulation of operator learning. Given a dataset

$$\mathcal{D} = \{(a^{(i)}, u^{(i)})\}_{i=1}^N, \qquad u^{(i)} \sim \mathcal{G}^\ddagger(a^{(i)}), \qquad (1)$$

the goal is to learn a parameterized operator $\mathcal{G}_\theta$ that approximates the ground truth mapping. Depending on whether the conditional distribution $\mathcal{G}^\ddagger(a)$ is degenerate or genuinely stochastic, we distinguish two regimes of operator learning.

**Problem 2.1** (Stochastic Problem). Given the dataset $\mathcal{D}$ in (1), the goal is to learn an approximation of the ground truth stochastic operator $\mathcal{G}_\theta \overset{d}{\approx} \mathcal{G}^\ddagger$.

This setting arises in stochastic dynamical systems with random forcing, unresolved microscale physics, and other regimes where output variability is intrinsic. Such stochasticity naturally appears in SPDE learning (Salvi et al., 2022; Chen et al., 2024; Shi et al., 2026) and in weather forecasting, where probabilistic prediction and ensemble uncertainty quantification are essential (Pathak et al., 2022; Price et al., 2025).

**Problem 2.2** (Deterministic Problem). Given the same dataset $\mathcal{D}$ in (1), there is a ground truth deterministic operator $\mathcal{G}^\dagger$, and the data are collected without randomness:

$$u^{(i)} = \mathcal{G}^\dagger(a^{(i)}) \qquad \forall i.$$

The goal is to learn an approximation $\mathcal{G}_\theta \approx \mathcal{G}^\dagger$.

This setting arises in surrogate modeling for deterministic physical systems, where one learns a fast approximation of a high-fidelity PDE solver mapping inputs to solution fields. Such surrogates enable rapid parameter sweeps, inverse problems, and design optimization (Kovachki et al., 2023; Azizzadenesheli et al., 2024). Even in this deterministic setting, uncertainty quantification is often desired by modeling a distribution over outputs to capture epistemic uncertainty from limited data or model misspecification.

Finally, although these problem classes are often studied using different model families, we adopt a unified perspective in which conditional diffusion models offer a flexible framework for treating deterministic and intrinsically stochastic operators within a common probabilistic formulation, while also providing a natural route to uncertainty quantification when needed.

## 2.2. Conditional Diffusion Models

In this subsection, we review conditional diffusion models from the perspective of operator learning, with the goal of modeling distributions over solution fields in stochastic operator learning. While several recent works formulate diffusion processes directly in infinite-dimensional function spaces (Lim et al., 2023; Kerrigan et al., 2023; 2024; Lim et al., 2025; Shi et al., 2025), our DLL framework instead operates on a compact finite-dimensional coefficient representation.

We slightly abuse notation and write the dataset as condition target pairs

$$\mathcal{D} = \{(c^{(i)}, x^{(i)})\}_{i=1}^N \qquad (2)$$

interchangeably with (1) for notational convenience. In (2), we assume the target output admits a finite-dimensional representation $x \in \mathbb{R}^{d_x}$ (e.g., latent variables or low-rank coefficient vectors).

We begin by recalling that diffusion models and flow matching models can be viewed under a unified generative framework (Lai et al., 2025; Gao et al., 2025). Throughout this paper, we refer to this unified family simply as *diffusion models*. The objective is the standard conditional generative modeling problem of learning the conditional law $p(x \mid c)$.

To this end, diffusion models construct a denoising mechanism that transports a simple noise distribution $p_{\text{noise}}$ to the target conditional distribution $p(\cdot \mid c)$. We introduce the forward noising process

$$x_t = a_t x + b_t \epsilon, \qquad 0 \le t \le 1, \qquad (3)$$

where $\epsilon \sim p_{\text{noise}}$ and $a_t, b_t$ define the noise schedule. Throughout this paper, we use the linear schedule $a_t = 1 - t$ and $b_t = t$. For notational convenience, we write $T = 1$ for the terminal diffusion time.

Given (3), the goal is to learn a reverse time dynamics that enables sampling from $p(x \mid c)$. Among several equivalent parameterizations, we focus on the *flow based* formulation via *velocity prediction* for clarity (Lipman et al., 2023; Liu et al., 2023). In this framework, there exists a velocity field $v(\cdot, t, c)$ such that the conditional densities $\rho_t(\cdot \mid c) = p(x_t \mid c)$ satisfy the continuity equation

$$\partial_t \rho_t + \nabla \cdot (\rho_t v) = 0, \qquad \rho_T(\cdot \mid c) = p_{\text{noise}}(\cdot).$$

Sampling is then performed by integrating the probability flow ODE backward in time:

$$\mathrm{d}x_t = v(x_t, t, c)\,\mathrm{d}t, \qquad x_T \sim p_{\text{noise}}.$$

The remaining task is to learn the velocity field from data. For the linear noising process (3), the *oracle* velocity along the coupling $(x, \epsilon)$ is given by $v^\star(x_t, t, c) = \dot{a}_t x + \dot{b}_t \epsilon$. Accordingly, we train a neural velocity field $v_\phi$ by minimizing the conditional velocity matching objective

$$\mathcal{L}_{\text{V}}(c) = \mathbb{E}_{x, \epsilon, t}\Big[\big\| v_\phi(x_t, t, c) - (\dot{a}_t x + \dot{b}_t \epsilon)\big\|_2^2\Big]. \quad (4)$$

To compare stochastic and deterministic conditional targets within a common distributional metric, we use the 2-Wasserstein distance between the learned distribution $\rho_0(\cdot \mid c)$ and the target distribution $p(\cdot \mid c)$. This choice is consistent with Wasserstein interpretations of diffusion training (Kwon et al., 2022) and is naturally aligned with velocity based transport stability, where errors in the learned velocity field translate into endpoint distributional errors (Benton et al., 2024).

**Proposition 2.3.** *Fix c. Under suitable regularity conditions, there exists a constant $C > 0$ such that*

$$\mathcal{W}_2(p(\cdot \mid c), \rho_0(\cdot \mid c)) \ \le\ C\sqrt{\mathcal{L}_{\text{V}}(c)}.$$

The proposition follows from stability estimates for the continuity equation under perturbations of the velocity field. Since the Wasserstein distance is defined on probability measures with finite second moment, it provides a common metric for both absolutely continuous target laws and singular measures such as Dirac deltas. Proposition 2.3 therefore supports using the same velocity matching objective $\mathcal{L}_{\text{V}}$ for both stochastic and deterministic conditional targets, corresponding to Problems 2.1 and 2.2. A proof and further discussion are provided in Appendix A.1.

It is also worth noting that (4) is minimized over the empirical dataset (2), rather than the full population distribution. Nevertheless, recent theoretical and empirical studies suggest that diffusion models can generalize beyond the training samples despite not driving the training loss to zero (Kadkhodaie et al., 2024; Bonnaire et al., 2025; Song et al., 2025). In practice, diffusion models often exhibit *underfitting* of (4), which can improve generalization.

## 3. Uncertainty Quantification and Probabilistic Surrogates

We examine how *conditional diffusion models* can serve as probabilistic surrogates for the stochastic and deterministic operator learning tasks in Problems 2.1 and 2.2. Table 1 summarizes the uncertainty representations considered in this work.

### 3.1. Uncertainty Quantification

We consider a probabilistic surrogate $\mathcal{G}_\theta$ for the ground truth stochastic operator $\mathcal{G}^\ddagger$. Here $\theta$ is an abstract uncertainty representation whose meaning depends on the model class. For

*Table 1.* Model comparison across uncertainty representations. Deterministic surrogates use point estimates, Bayesian models infer parameter uncertainty, and diffusion based surrogates learn conditional generative models for the output distribution $\mathcal{G}^{\ddagger}(a)$.

| Methods | $\theta$ | Estimates | $\mathcal{G}_{\theta}(a)$ |
|---------|----------|-----------|----------------------------|
| NN | NN Param | $\theta \mid \mathcal{D}$ | $\approx \mathcal{G}^{\dagger}(a)$ |
| BNN | NN Param | $p(\theta \mid \mathcal{D})$ | $\overset{p}{\approx} \mathcal{G}^{\dagger}(a)$ |
| DM | Data Vector | $p(\theta \mid \mathcal{D}, a)$ | $\overset{d}{\approx} \mathcal{G}^{\ddagger}(a)$ |
| LDM | Latent Vector | $p(\theta \mid \mathcal{D}, a)$ | $\overset{d}{\approx} \mathcal{G}^{\ddagger}(a)$ |
| DLL | Coeff. Vector | $p(\theta \mid \mathcal{D}, a)$ | $\overset{d}{\approx} \mathcal{G}^{\ddagger}(a)$ |

deterministic and Bayesian neural network surrogates, $\theta$ denotes network parameters or their posterior distribution. For diffusion based surrogates, $\theta$ denotes the random output representation being modeled, such as a data vector, latent vector, or coefficient vector. Thus, depending on the formulation, $\theta$ may represent a point estimate, a parameter posterior, or a distribution over output representations.

For a fixed input $a$, the total distributional error can be decomposed as

$$\underbrace{\mathcal{W}_2\big(\mathcal{G}^{\ddagger}(a),\, \mathcal{G}_{\theta}(a)\big)}_{\text{total error}} \leq \underbrace{\mathcal{W}_2\big(\mathcal{G}^{\ddagger}(a),\, \mathcal{G}_{\theta^{\star}}(a)\big)}_{\text{model misspecification}}$$
$$+ \underbrace{\mathcal{W}_2\big(\mathcal{G}_{\theta^{\star}}(a),\, \mathcal{G}_{\theta}(a)\big)}_{\text{epistemic uncertainty}},$$

where $\theta^{\star}$ is an oracle estimator associated with the hypothesis class and optimization procedure. The first term reflects the intrinsic limitation of the chosen surrogate family, while the second term arises from finite data, optimization error, and model uncertainty. In addition, $\mathcal{G}^{\ddagger}$ may exhibit *aleatoric uncertainty*, namely irreducible randomness in the conditional output distribution. The goal of UQ is therefore to capture both aleatoric variability and epistemic uncertainty in a predictive distribution over functions.

### 3.2. Classical Probabilistic Surrogates

Classical probabilistic surrogates often adopt Bayesian or approximate Bayesian perspectives, where uncertainty is represented through a posterior distribution over parameters, $p(\theta \mid \mathcal{D})$. Under a specified prior and likelihood, prediction is performed through the posterior predictive distribution

$$p(u \mid a, \mathcal{D}) = \int p(u \mid a, \theta)\, p(\theta \mid \mathcal{D})\, d\theta.$$

Exact Bayesian inference is generally intractable for modern neural networks, so scalable approximations such as Monte Carlo dropout (Gal & Ghahramani, 2016) and deep ensembles (Lakshminarayanan et al., 2017) are commonly used.

These approaches primarily represent epistemic uncertainty through parameter uncertainty, while aleatoric uncertainty is typically mediated by the assumed likelihood model. As a result, the form of intrinsic variability is often prescribed rather than learned directly as a rich conditional output distribution. Moreover, the resulting uncertainty estimates may require post hoc calibration (Guo et al., 2017; Kuleshov et al., 2018). These limitations are especially relevant in stochastic PDE settings, where the conditional solution distribution can be spatially structured and strongly non Gaussian.

### 3.3. Conditional Diffusion Models as Probabilistic Surrogates

In this work, conditional diffusion models directly learn a predictive distribution over output representations conditioned on the input, denoted by $p(\theta \mid \mathcal{D}, a)$. This contrasts with Bayesian neural operator approaches, which infer a posterior over model parameters and obtain predictive uncertainty through the resulting posterior predictive distribution. Conditional generative models instead learn a family of output distributions indexed by $a$, making them naturally suited to Problem 2.1.

In deterministic or nearly deterministic settings, diffusion underfitting may also provide a heuristic indication of epistemic uncertainty. If the underlying solution operator is deterministic, the conditional output law should ideally concentrate around a single solution for each condition $c$. Residual variability in the learned distribution is then more naturally attributed to limited data, model approximation error, or incomplete optimization rather than irreducible aleatoric randomness. In this sense, a nonzero $\mathcal{L}_{\mathrm{V}}(c)$ can be viewed as a qualitative indicator of residual uncertainty associated with limited information in $\mathcal{D}$.

Finally, the learned conditional generator satisfies a useful stability property. If the learned velocity field $v_{\phi}$ is Lipschitz continuous in the condition $c$, then the generated conditional distributions vary continuously with $c$ in $\mathcal{W}_2$. This shows that the learned output law changes in a controlled manner under small perturbations of the input condition. We emphasize, however, that this regularity property is not a calibrated epistemic uncertainty guarantee. A precise statement is provided in Appendix A.1.

## 4. Diffusion Last Layer Neural Operators

In this section, we introduce the proposed DLL and summarize its training and inference pipeline, as illustrated in Fig. 2. We first describe the operator encoder and then present the conditional diffusion model defined in coefficient space.

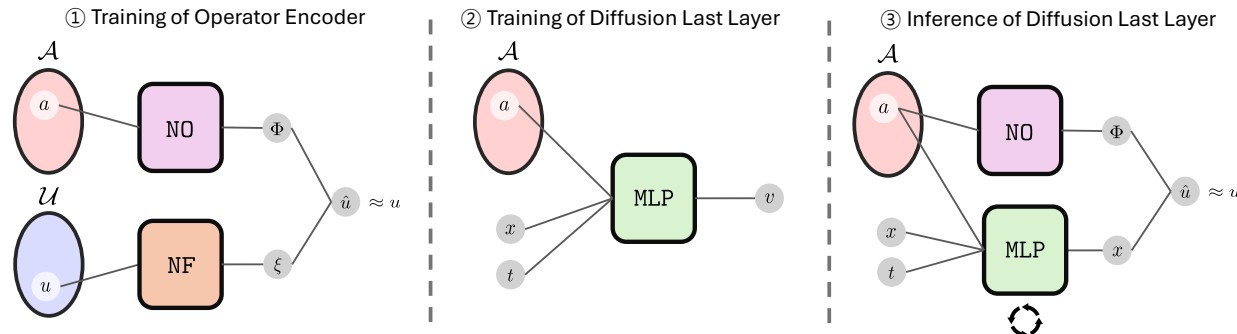

*Figure 2.* **Training and inference pipeline for DLL. (1) Operator encoder.** A `NO` backbone produces basis functions $\Phi(a)$, and a `NF` maps targets to coefficients $\xi = \text{NF}(u)$, yielding $\hat{u} = \xi^\top \Phi(a)$. **(2) DLL training.** With the encoder frozen, we train a conditional diffusion model in coefficient space using an MLP velocity model conditioned on features of $a$. **(3) DLL inference.** For a new input $a$, we sample coefficients $\theta \sim p(\theta \mid \mathcal{D}, a)$ by integrating the learned probability flow ODE and decode $\hat{u} = \theta^\top \Phi(a)$.

### 4.1. Operator Encoder

Given the dataset $\mathcal{D}$ in (1), we learn an *operator encoder* consisting of two components: an input dependent basis generator and an output coefficient encoder. The basis generator is a neural operator $\text{NO}_\psi$ that maps the input field $a$ to basis functions $\Phi_a$, while the coefficient encoder $\text{NF}_\varphi$ maps the target field $u$ to coefficients $\xi$. Concretely, we represent $u$ by a rank-$r$ expansion

$$u \approx \hat{u} := \sum_{k=1}^{r} \xi_k \, \phi_k(a) = \xi^\top \Phi_a,$$

where $\Phi_a = (\phi_1(a), \ldots, \phi_r(a)) \in \mathcal{U}^r$ denotes an input dependent collection of basis functions produced by the neural operator backbone, and $\xi = (\xi_1, \ldots, \xi_r) \in \mathbb{R}^r$ is an instance specific coefficient vector.

In our implementation, the coefficient vector is produced by a neural functional encoder $\text{NF}_\varphi$, namely $\xi = \text{NF}_\varphi(u)$, while the input dependent basis is generated by a neural operator $\text{NO}_\psi$, namely $\Phi_a = \text{NO}_\psi(a)$. With the convention that the product between a coefficient vector and a vector of functions denotes the corresponding linear combination, the reconstruction is given by

$$\hat{u} = \text{NF}_\varphi(u)^\top \text{NO}_\psi(a).$$

This construction provides a compact conditional representation of the target field through the coefficient vector $\xi$, which is used as the latent variable for diffusion modeling in the next subsection.

We train the operator encoder by minimizing the reconstruction loss

$$\mathcal{L}_{\text{OE}} = \mathbb{E}_{(a,u)\sim\mathcal{D}} \big\| u - \text{NF}_\varphi(u)^\top \text{NO}_\psi(a) \big\|_2^2, \quad (5)$$

which encourages the learned coefficients and input dependent basis functions to provide an accurate rank-$r$ approximation of $u$. Intuitively, $\text{NO}_\psi(a)$ produces a basis adapted to the input condition $a$, while $\text{NF}_\varphi(u)$ extracts the corresponding instance specific coefficients, so that their combination reconstructs the target field with small mean squared error.

Under idealized approximation assumptions, minimizing (5) is related to the optimal rank-$r$ reconstruction of the conditional output field.

**Proposition 4.1** (Optimal rank-$r$ reconstruction)**.** *Fix $r \in \mathbb{N}$ and $a \in \mathcal{A}$. Under assumptions stated in Appendix A.2, minimizing $\mathcal{L}_{\text{OE}}$ in (5) recovers the optimal rank-$r$ reconstruction of $u$ conditioned on $a$, in the sense that*

$$\inf \mathcal{L}_{\text{OE}}(a) = \inf_{\dim(S)=r} \mathbb{E}\big[\|u - P_S u\|^2 \mid a\big].$$

*The minimizing subspace is spanned by the leading $r$ eigenfunctions of the conditional second moment operator. Equivalently, the learned basis can be interpreted as an input dependent uncentered Karhunen–Loève subspace.*

A precise statement and proof are provided in Appendix A.2. As a result, the operator encoder compresses the high-dimensional output field into a compact coefficient vector $\xi \in \mathbb{R}^r$, in contrast to conventional latent diffusion models that rely on discrete autoencoder latents.

Compared with a standard autoencoder, the operator encoder can improve reconstruction efficiency at a fixed latent dimension by using an input dependent basis $\Phi_a = \text{NO}_\psi(a)$. Since the reconstruction subspace adapts to the condition $a$, more of the output structure can be represented through $\Phi_a$, reducing the burden on the low-dimensional coefficient vector $\xi$ and mitigating reconstruction error caused by the bottleneck.

### 4.2. Diffusion Last Layer

Given the trained operator encoder, we construct the latent dataset in (2) by setting $c^{(i)} = a^{(i)}$ and $x^{(i)} = \text{NF}_\varphi(u^{(i)})$. This reduces conditional generation to modeling a distribu-

tion over the finite-dimensional coefficient space $\mathbb{R}^r$, rather than directly over the output function space $\mathcal{U}$.

We train the diffusion last layer by minimizing the velocity loss $\mathcal{L}_V$ in (4). In contrast to pixel space diffusion models, DLL performs both training and inference in the low-dimensional coefficient space, which substantially reduces computational cost and improves sampling efficiency. Since the latent variable $x \in \mathbb{R}^r$ is finite-dimensional, we parameterize the conditional velocity field using an MLP, a standard architecture for generative modeling in vector spaces (Kotelnikov et al., 2023; Li et al., 2024).

The preceding theoretical results offer a useful interpretation of this construction. Proposition 2.3 indicates that, under the stated assumptions, minimizing $\mathcal{L}_V$ controls the discrepancy between the learned conditional latent distribution and the target conditional law of $x$ in Wasserstein distance. Proposition 4.1 suggests that the operator encoder can be viewed as learning an input-adaptive rank-$r$ uncentered Karhunen–Loève-type representation of the output field. In this sense, DLL models the conditional output distribution by learning a diffusion model over the corresponding coefficient space and decoding sampled coefficients through the input dependent basis $\mathtt{NO}_\psi(a)$.

# 5. Experiments

We evaluate DLL on both stochastic and deterministic benchmarks. Unless otherwise stated, all main experiments use FNO (Li et al., 2020) as the backbone neural operator $\mathtt{NO}$, latent dimension $r = 64$, and NFE = 10 sampling steps. The neural functional encoder $\mathtt{NF}$ uses the same FNO backbone architecture, followed by global average pooling, to obtain the latent coefficients. Further implementation details are provided in Appendix B. Appendix C reports ablation studies on the latent dimension, the number of sampling NFEs, the training dataset size, and backbone compatibility with DeepONet.

## 5.1. Choice of Baselines

We compare DLL against five baselines covering deterministic prediction, approximate Bayesian uncertainty estimation, and conditional generative modeling. As operator-based references, we include FNO (Li et al., 2020), FNO-d, which applies Monte Carlo dropout at inference time, and PNO (Bülte et al., 2025), implemented as the reparameterized variant of the probabilistic neural operator. In our implementation, PNO outputs pointwise Gaussian predictive distributions through mean and standard-deviation heads and draws samples using the reparameterization trick. We further evaluate two grid-based generative baselines: pixel space diffusion (DM) and latent diffusion (LDM). Since all benchmarks are posed on regular grids, these discretized diffusion baselines

allow direct comparison in the same data representation. Additional architectural details are provided in Appendix B.1.

## 5.2. Stochastic Operator Learning

We evaluate whether probabilistic surrogates can capture the aleatoric uncertainty arising from stochastic PDE data, where each input corresponds to a conditional distribution over output fields. For each stochastic system, we generate 10,000 training pairs and evaluate on 32 test inputs with 64 output realizations per input. The one-dimensional Burgers fields are represented as $u \in \mathbb{R}^{256}$, while the two-dimensional Darcy fields are represented as $u \in \mathbb{R}^{128 \times 128}$. Full dataset generation details are provided in Appendix B.3.

**Stochastic Burgers' Equation.** We consider the one-dimensional viscous stochastic Burgers' equation on the periodic domain $x \in [0, 2\pi]$:

$$\mathrm{d}u = \left(-\frac{1}{2}\partial_x(u^2) + \nu\,\partial_{xx}u\right)\mathrm{d}t + \sum_{j \in \mathcal{J}} w_j \cos(jx)\,\mathrm{d}W_t^j,$$

where $\nu > 0$ is the viscosity, $\sigma > 0$ is the overall noise scale, $\mathcal{J}$ is a finite set of forcing modes, $w_j$ are mode weights, and $\{W_t^j\}_{j \in \mathcal{J}}$ are independent standard Brownian motions. Given an initial condition $u_0 = u(\cdot, 0)$, randomness in the Brownian forcing induces a conditional distribution over terminal time solutions $u(\cdot, T)$.

**Stochastic Darcy Flow.** We consider Darcy flow on $\Omega = (0, 1)^2$ with zero Dirichlet boundary conditions:

$$-\nabla \cdot \big(a(x)\nabla u(x)\big) = f(x), \qquad x \in \Omega,$$
$$u(x) = 0, \qquad x \in \partial\Omega,$$

where $a(x)$ is the input permeability field and $u(x)$ is the pressure field. Aleatoric uncertainty arises from the random source

$$f(x) = \sigma_{\mathrm{ln}} \exp\big(G_{\mathrm{ln}}(x)\big) + \sigma_{\mathrm{gp}} G_{\mathrm{gp}}(x),$$

where $\sigma_{\mathrm{ln}}, \sigma_{\mathrm{gp}} > 0$, and $G_{\mathrm{ln}}$ and $G_{\mathrm{gp}}$ are independent mean-zero Gaussian random fields. Given a permeability field $a$, the random source induces a conditional distribution over pressure fields $u$.

*Table 2.* **Stochastic Burgers' equation.** Results on distributional and moment metrics. Lower is better. Best and second-best results are highlighted in **bold** and underline, respectively.

| Method | ED $\downarrow$ | SWD $\downarrow$ | NRMSE$_m$ $\downarrow$ | NRMSE$_s$ $\downarrow$ |
|--------|------|-------|------|------|
| FNO | 6.491 | 0.426 | **0.146** | 1.000 |
| FNO-d | 6.075 | 0.387 | 0.527 | 0.755 |
| PNO | 1.766 | 0.253 | 0.215 | 0.457 |
| DM | 1.355 | 0.239 | 0.258 | 0.323 |
| LDM | 1.373 | 0.249 | 0.280 | 0.297 |
| DLL | **1.285** | **0.213** | 0.252 | **0.289** |

*Table 3.* **Stochastic Darcy flow.** Results on distributional and moment metrics. Lower is better. Best and second best results are highlighted in **bold** and underline, respectively.

| Method | ED ↓ | SWD ↓ | NRMSE$_m$ ↓ | NRMSE$_s$ ↓ |
|---|---|---|---|---|
| FNO | 1.463 | 0.015 | **0.253** | 1.000 |
| FNO-d | 1.320 | 0.014 | 0.289 | 0.962 |
| PNO | 0.305 | 0.007 | 0.388 | 0.285 |
| DM | 0.269 | 0.007 | 0.353 | 0.360 |
| LDM | 0.368 | 0.007 | 0.610 | **0.268** |
| DLL | **0.227** | 0.007 | 0.355 | 0.357 |

**Results.** Across both stochastic benchmarks (Tables 2 and 3), FNO achieves the lowest mean error but does not capture output variability, leading to large ED, SWD, and NRMSE$_s$. FNO-d improves uncertainty estimates only modestly, while PNO provides a stronger probabilistic baseline but remains less competitive on distributional metrics. The generative baselines substantially reduce ED and SWD. Among them, DLL achieves the lowest ED on both benchmarks and the lowest SWD on stochastic Burgers, while remaining competitive on stochastic Darcy. Metric definitions and qualitative results are provided in Appendices B.4 and D.1, respectively.

## 5.3. Autoregressive Rollout Stability

We next evaluate long horizon autoregressive stability on deterministic chaotic dynamics. Following APEBench (Koehler et al., 2024), models are trained for one step prediction and evaluated by autoregressive rollouts, where predictions are recursively fed back as inputs. This setting probes error accumulation and compounding drift over time. For both benchmarks, models are trained on trajectory segments of length 50 and evaluated on rollouts of length 100. Dataset generation details are provided in Appendix B.3.

**Kuramoto–Sivashinsky Equation.** We consider the one-dimensional Kuramoto–Sivashinsky equation on the periodic domain $x \in [0, L]$:

$$\partial_t u + u\,\partial_x u + \partial_{xx} u + \partial_{xxxx} u = 0.$$

The state is represented as $u \in \mathbb{R}^{256}$ on a uniform grid. The autoregressive task is to learn the discrete time flow map from $u(\cdot, t)$ to $u(\cdot, t + \Delta t)$ with $\Delta t = 1$.

**Kolmogorov Flow.** We consider two-dimensional Kolmogorov flow on the periodic domain $\Omega = (0, 2\pi)^2$, modeled by the incompressible Navier–Stokes equations in vor-

*Table 4.* **Kuramoto–Sivashinsky equation.** Autoregressive rollout performance. Lower is better for NRMSE and CRPS, and SSR is optimal when close to 1. Best and second best results are highlighted in **bold** and underline, respectively.

| Method | NRMSE ↓ | CRPS ↓ | SSR → 1 |
|---|---|---|---|
| FNO | 0.404 | – | – |
| FNO-d | 0.384 | 0.523 | **0.975** |
| PNO | 0.354 | 0.514 | 0.550 |
| DM | 0.395 | 0.545 | 0.961 |
| LDM | 0.576 | 0.878 | 0.802 |
| DLL | **0.343** | **0.470** | 0.949 |

*Table 5.* **Kolmogorov flow.** Autoregressive rollout performance. Lower is better for NRMSE and CRPS, and SSR is optimal when close to 1. Best and second best results are highlighted in **bold** and underline, respectively.

| Method | NRMSE ↓ | CRPS ↓ | SSR → 1 |
|---|---|---|---|
| FNO | 0.528 | – | – |
| FNO-d | 0.463 | 0.912 | 0.546 |
| PNO | 0.492 | 1.119 | 0.167 |
| DM | **0.369** | **0.692** | 0.601 |
| LDM | 0.615 | 1.232 | 0.548 |
| DLL | 0.426 | 0.822 | **0.620** |

ticity form:

$$\partial_t \omega + \mathbf{u} \cdot \nabla \omega = \nu \Delta \omega - \alpha \omega + F(x),$$
$$\Delta \psi = \omega,$$
$$\mathbf{u} = \nabla^\perp \psi,$$

where $\omega$ is the vorticity, $\psi$ is the stream function, $\nu > 0$ is the viscosity, $\alpha \geq 0$ is the linear drag, and $F$ is a single mode Kolmogorov forcing. The state $\omega(\cdot, t)$ is represented on a uniform grid as $\omega \in \mathbb{R}^{128 \times 128}$. The autoregressive task is to learn the discrete time flow map from $\omega(\cdot, t)$ to $\omega(\cdot, t + \Delta t)$ with $\Delta t = 0.25$.

**Results.** Tables 4 and 5 summarize autoregressive rollout performance. On KS, DLL achieves the best NRMSE and CRPS while maintaining SSR close to one, indicating improved rollout accuracy with reasonable uncertainty estimates. On Kolmogorov flow, pixel space diffusion (DM) gives the best NRMSE and CRPS, while DLL improves over the deterministic FNO backbone and attains the best SSR. We attribute this gap partly to the stronger spatial inductive bias of the U-Net used by DM and partly to the fact that DLL operates through a low-dimensional coefficient space attached to the FNO backbone; thus, its performance ceiling can depend on the quality of the underlying backbone. Overall, the results suggest that DLL can improve autoregressive stability of neural operator backbones, although pixel space diffusion may remain advantageous for highly complex dynamics. Metric definitions and qualitative results are provided in Appendices B.4 and D.2, respectively.

*Table 6.* **Reconstruction property (1D).** Autoencoder versus operator encoder. Lower NRMSE indicates better reconstruction.

| Encoder | Comp. ↑ | Burgers NRMSE ↓ | KS NRMSE ↓ |
|---------|---------|-----------------|------------|
| AE | $\times 2$ | $\mathbf{1.98 \times 10^{-3}}$ | $7.75 \times 10^{-4}$ |
| OE | $\times \mathbf{4}$ | $4.13 \times 10^{-2}$ | $\mathbf{2.45 \times 10^{-4}}$ |

*Table 7.* **Reconstruction property (2D).** Autoencoder versus operator encoder. Lower NRMSE indicates better reconstruction.

| Encoder | Comp. ↑ | Darcy NRMSE ↓ | Kolmogorov NRMSE ↓ |
|---------|---------|---------------|--------------------|
| AE | $\times 16$ | $\mathbf{1.05 \times 10^{-2}}$ | $3.35 \times 10^{-3}$ |
| OE | $\times \mathbf{256}$ | $4.11 \times 10^{-2}$ | $\mathbf{3.30 \times 10^{-3}}$ |

## 5.4. Reconstruction Property

We finally examine whether the proposed operator encoder provides an effective compressed representation of the output field, since this representation is the latent space in which DLL performs diffusion modeling. Tables 6 and 7 show that operator encoder (OE) reconstructs deterministic benchmarks (KS and Kolmogorov flow) more accurately than autoencoder (AE), while AE performs better on stochastic benchmarks (Burgers and Darcy). Notably, OE achieves lower error in the deterministic setting despite substantially higher compression ratios, suggesting that operator conditioned features capture the dominant low-dimensional structure of the solution manifold.

## 6. Related Work and Discussion

**UQ in Operator Learning.** UQ for operator learning has been studied from several perspectives. One line targets *epistemic* uncertainty through Bayesian neural operators and related approximate Bayesian formulations over neural operator components (Lin et al., 2023; Weber et al., 2024; Magnani et al., 2025a;b), as well as input perturbation or ensemble prediction mechanisms (Pathak et al., 2022). Recent diffusion inspired neural operator parametrizations have also been explored for efficient Bayesian uncertainty estimation in Fourier neural operators (Matveev et al., 2025). Another line models uncertainty directly in output function space, such as probabilistic neural operators trained with proper scoring rules (Bülte et al., 2025). In contrast, DLL learns a flexible conditional generative model in a low-dimensional coefficient space defined by a neural operator backbone, capturing aleatoric variability in stochastic problems and providing useful predictive spreads in deterministic rollouts. Separately, conformal methods provide distribution-free calibrated uncertainty sets for neural operators (Ma et al., 2024; Millard et al., 2025); extending such guarantees to DLL is an important direction for future work.

**Generative Models for Physics.** The use of generative models in the physical sciences has grown rapidly in recent years (Shu et al., 2023; Shysheya et al., 2024; Kohl et al., 2024; Zhou et al., 2025a;b; Li et al., 2025; Rozet et al., 2025). Most existing approaches adapt latent space or pixel space generative models to discretized simulation data, often using U-Nets, diffusion transformers, or neural fields. For example, neural field latent diffusion has been used to model spatiotemporal turbulence (Du et al., 2024), while wavelet based diffusion architectures have been developed for generative PDE simulation and control (Hu et al., 2025). DLL instead takes an operator learning perspective by adding a conditional generative head to a neural operator backbone and sampling in the final coefficient space. Another important direction is to extend DLL to Bayesian inverse problems, where conditional generative operator models could help represent learned posterior surrogates when combined with suitable backbones (Rozet & Louppe, 2023; Huang et al., 2024). Combining DLL with emerging pretrained scientific models (Hao et al., 2024; Herde et al., 2024) is a promising future direction, but would require appropriate training and adaptation strategies.

**Generative Models in Function Spaces.** A closely related line of work formulates diffusion and flow matching directly over *function-valued* random variables to obtain discretization-robust generative models, using Gaussian-process, Hilbert-space, or function-space score-matching constructions (Lim et al., 2023; Kerrigan et al., 2023; 2024; Lim et al., 2025; Shi et al., 2025). Spectral diffusion processes are particularly related to DLL because they model functional data through coefficients in a fixed spectral or KL-type basis (Phillips et al., 2022). DLL shares the coefficient space viewpoint, but learns an input dependent basis through a neural operator backbone and performs conditional flow matching over the resulting coefficients. Thus, DLL can be viewed as a conditional function space generative model with a coefficient parameterization learned through a neural operator, rather than a fully infinite-dimensional diffusion model.

**Neural Processes.** Neural processes provide a probabilistic framework for learning distributions over functions from context observations and predicting at target locations (Garnelo et al., 2018a;b). Subsequent variants improve predictive correlations, equivariance, scalability, and spatiotemporal modeling through Gaussian, convolutional, diffusion-based, transformer-based, and spectral constructions (Bruinsma et al., 2021; Gordon et al., 2020; Dutordoir et al., 2023; Ashman et al., 2025; Mohseni & Duffield, 2025). This family of methods is closely related to DLL in its emphasis on probabilistic function modeling and uncertainty-aware prediction. DLL brings a similar perspective to operator learning by using a coefficient representation and learning a

conditional generative model over output functions. While neural processes are typically formulated around context-to-target prediction, DLL focuses on operator learning, where the conditioning variable is an input function and the target is the corresponding output field distribution.

**Weight Space Uncertainty.** A complementary route to uncertainty quantification models randomness in parameter space. Bayesian neural networks infer a posterior over weights, often via variational methods, to capture epistemic uncertainty (Blundell et al., 2015). For scalability, Bayesian last-layer models restrict posterior inference to the final layer while learning a deterministic feature extractor (Kristiadi et al., 2020; Watson et al., 2021; Harrison et al., 2024). More recently, diffusion models have been used to learn expressive distributions over network weights and enable sampling in weight space (Erkoç et al., 2023; Xie et al., 2024). In contrast, DLL keeps the backbone deterministic and models uncertainty in a low-dimensional output representation, targeting aleatoric variability while remaining complementary to weight-space approaches.

## 7. Conclusion

We introduced DLL, a modular probabilistic output head for neural operator backbones. DLL learns an operator encoder that represents target fields through input dependent basis functions and compact coefficients, and then trains a conditional diffusion model in this coefficient space. This design provides an efficient way to model conditional distributions over solution fields while retaining the structural advantages of operator learning.

Across stochastic operator learning benchmarks, DLL improves distributional fidelity over deterministic and probabilistic operator baselines and remains competitive with grid based diffusion baselines. On deterministic chaotic systems, DLL improves the rollout stability of the underlying neural operator backbone and provides informative uncertainty estimates under compounding autoregressive errors. These results suggest that diffusion modeling in learned coefficient spaces is a practical route to uncertainty aware neural operators. Future work includes principled calibration with coverage guarantees, stronger operator backbones, and extensions to inverse problems and irregular geometries.

## Impact Statement

This paper presents work whose goal is to advance the field of machine learning. There are many potential societal consequences of our work, none of which we feel must be specifically highlighted here.

## Acknowledgment

This work was supported by Institute of Information & communications Technology Planning & Evaluation (IITP) grant funded by the Korea government(MSIT) (RS-2022-00143911, AI Excellence Global Innovative Leader Education Program)

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

# A. Theoretical Analysis

## A.1. UQ with Diffusion Models

In this section, we provide a concise stability argument connecting velocity matching training to endpoint error in Wasserstein distance. Since our setting includes both genuinely stochastic targets and deterministic point-mass targets, $\mathcal{W}_2$ provides a common metric for probability measures with finite second moments.

**Assumption A.1** (Regularity of probability paths). Fix a condition $c$. Let $\rho_t^\star(\cdot \mid c)$ denote the target probability path defined by the noising process (3), with

$$\rho_0^\star(\cdot \mid c) = p(\cdot \mid c), \qquad \rho_T^\star(\cdot \mid c) = p_{\text{noise}}.$$

Let $\rho_t(\cdot \mid c)$ denote the probability path generated by the learned velocity field $v_\phi(\cdot, t, c)$, with the same terminal condition $\rho_T(\cdot \mid c) = p_{\text{noise}}$. We assume that both paths are narrowly continuous probability measures on $\mathbb{R}^d$ with finite second moments, solve their corresponding continuity equations, and have finite kinetic energy. We also assume that $v_\phi(\cdot, t, c)$ is $L(t)$-Lipschitz in the state variable for a.e. $t \in [0, T]$, where $L \in L^1(0, T)$.

**Proposition A.2** (Endpoint stability). *Under Assumption A.1, let $v^\star(\cdot, t, c)$ denote the marginal velocity field of the target path $\rho_t^\star(\cdot \mid c)$, and define*

$$\Lambda(t) := \int_0^t L(s)\, ds.$$

*Then*

$$\mathcal{W}_2(\rho_0^\star(\cdot \mid c), \rho_0(\cdot \mid c)) \leq \int_0^T \exp\big(\Lambda(t)\big) \big(\mathbb{E}_{x_t \sim \rho_t^\star(\cdot \mid c)}\big[\|v_\phi(x_t, t, c) - v^\star(x_t, t, c)\|^2\big]\big)^{1/2} dt.$$

*Consequently,*

$$\mathcal{W}_2(\rho_0^\star(\cdot \mid c), \rho_0(\cdot \mid c)) \leq C_L \left( \int_0^T \mathbb{E}_{x_t \sim \rho_t^\star(\cdot \mid c)}\big[\|v_\phi(x_t, t, c) - v^\star(x_t, t, c)\|^2\big] dt \right)^{1/2},$$

*where*

$$C_L := \left( \int_0^T \exp\big(2\Lambda(t)\big) dt \right)^{1/2}.$$

*Proof.* We apply a standard stability estimate for the continuity equation in $\mathcal{W}_2$; see, e.g., Ambrosio et al. (2005); Benton et al. (2024). If one velocity field is Lipschitz in the state variable, then the distance between two solutions is controlled by the accumulated $L^2$ discrepancy between the two velocity fields along the reference path, up to a Grönwall factor.

Since the target and learned paths share the terminal condition at $t = T$, we apply this estimate in reversed time, with the learned velocity field $v_\phi$ as the Lipschitz field and the target path $\rho_t^\star$ as the reference path. This gives

$$\mathcal{W}_2(\rho_0^\star(\cdot \mid c), \rho_0(\cdot \mid c)) \leq \int_0^T \exp\big(\Lambda(t)\big) \|v_\phi(\cdot, t, c) - v^\star(\cdot, t, c)\|_{L^2(\rho_t^\star)} dt,$$

which proves the first inequality. The second inequality follows from Cauchy–Schwarz in time. $\qquad \square$

*Proof of Proposition 2.3.* For the linear noising process $x_t = a_t x + b_t \epsilon$, the sample-level velocity target is

$$\dot{x}_t = \dot{a}_t x + \dot{b}_t \epsilon.$$

The corresponding marginal velocity field is

$$v^\star(x_t, t, c) = \mathbb{E}\big[\dot{a}_t x + \dot{b}_t \epsilon \mid x_t, c\big].$$

By the projection property of conditional expectation,

$$\mathbb{E}_{x_t \sim \rho_t^\star(\cdot|c)}\left[\|v_\phi(x_t, t, c) - v^\star(x_t, t, c)\|^2\right] \leq \mathbb{E}\left[\|v_\phi(x_t, t, c) - (\dot{a}_t x + \dot{b}_t \epsilon)\|^2 \mid c\right].$$

Combining this inequality with Proposition A.2 and the definition of $\mathcal{L}_V(c)$ in (4) yields

$$\mathcal{W}_2(p(\cdot \mid c), \rho_0(\cdot \mid c)) \leq C\sqrt{\mathcal{L}_V(c)}$$

for a constant $C > 0$ depending on the time horizon, the time-sampling normalization, and the Lipschitz regularity. This proves the claim. $\qquad\square$

We also record a simple stability property with respect to the conditioning variable.

**Assumption A.3** (Lipschitz continuity in the condition). There exists $L_c > 0$ such that for all $c_1, c_2$, a.e. $t \in [0, T]$, and all $x \in \mathbb{R}^d$,

$$\|v_\phi(x, t, c_1) - v_\phi(x, t, c_2)\| \leq L_c \|c_1 - c_2\|.$$

**Proposition A.4** (Conditional stability). *Assume that the learned paths for $c_1$ and $c_2$ satisfy Assumption A.1 with the same state-Lipschitz function $L(t)$, and suppose Assumption A.3 holds. Then*

$$\mathcal{W}_2(\rho_0(\cdot \mid c_1), \rho_0(\cdot \mid c_2)) \leq L_c \left(\int_0^T \exp(\Lambda(t)) dt\right) \|c_1 - c_2\|.$$

*Proof.* Apply the same endpoint stability argument as in Proposition A.2 to the two learned paths generated under $c_1$ and $c_2$. These paths share the same terminal distribution $p_{\text{noise}}$. By Assumption A.3,

$$\|v_\phi(\cdot, t, c_1) - v_\phi(\cdot, t, c_2)\|_{L^2(\rho_t(\cdot|c_2))} \leq L_c \|c_1 - c_2\|.$$

Substituting this bound into the endpoint stability estimate gives the result. $\qquad\square$

Proposition A.4 shows that small perturbations of the condition lead to controlled perturbations of the generated law. This should be interpreted as a regularity property of the learned conditional generator, rather than as a calibrated epistemic uncertainty guarantee.

## A.2. Diffusion Last Layer

We formalize the operator encoder reconstruction problem at a fixed condition $a$ in an abstract Hilbert space setting. This isolates the rank-$r$ approximation property independently of any particular parameterization.

**Assumption A.5** (Hilbert setting and conditional second moments). Let $\mathcal{U}$ be a separable Hilbert space with inner product $\langle \cdot, \cdot \rangle$ and norm $\| \cdot \|$. Fix $a \in \mathcal{A}$ and let $u$ be a $\mathcal{U}$-valued random element under the conditional law $\mathbb{P}(\cdot \mid a)$ such that $\mathbb{E}[\|u\|^2 \mid a] < \infty$. Let $\Phi(a) = (\phi_1(a), \ldots, \phi_r(a)) \in \mathcal{U}^r$ define the subspace $S(a) := \text{span}\{\phi_1(a), \ldots, \phi_r(a)\} \subset \mathcal{U}$, and let $P_{S(a)}$ denote the orthogonal projector onto $S(a)$.

**Lemma A.6** (Projection decomposition). *Under Assumption A.5, for any measurable map $\hat{u} : \mathcal{U} \to S(a)$,*

$$\|u - \hat{u}(u)\|^2 = \|u - P_{S(a)}u\|^2 + \|P_{S(a)}u - \hat{u}(u)\|^2, \qquad \mathbb{P}(\cdot \mid a)\text{-a.s.}$$

*Consequently,*

$$\mathbb{E}\left[\|u - \hat{u}(u)\|^2 \mid a\right] = \mathbb{E}\left[\|u - P_{S(a)}u\|^2 \mid a\right] + \mathbb{E}\left[\|P_{S(a)}u - \hat{u}(u)\|^2 \mid a\right].$$

*Proof.* Since $u - P_{S(a)}u \perp S(a)$ and $P_{S(a)}u - \hat{u}(u) \in S(a)$, the cross term vanishes. Expanding the squared norm and taking conditional expectation gives the result. $\qquad\square$

**Proposition A.7** (Fixed basis: optimal encoder equals orthogonal projection). *Fix $a$ and $\Phi(a)$ as in Assumption A.5. Consider reconstructions of the form*

$$\hat{u}_\xi(u) = \sum_{k=1}^r \xi_k(u)\phi_k(a) \in S(a),$$

*where $\xi : \mathcal{U} \to \mathbb{R}^r$ is measurable. Define*

$$\mathcal{L}(\xi; a, \Phi) := \mathbb{E}\big[\|u - \hat{u}_\xi(u)\|^2 \mid a\big].$$

*Then*

$$\inf_\xi \mathcal{L}(\xi; a, \Phi) = \mathbb{E}\big[\|u - P_{S(a)}u\|^2 \mid a\big],$$

*and any minimizer satisfies $\hat{u}_{\xi^\star}(u) = P_{S(a)}u \ \mathbb{P}(\cdot \mid a)$-a.s. If the Gram matrix $G \in \mathbb{R}^{r \times r}$, $G_{ij} = \langle \phi_i(a), \phi_j(a) \rangle$, is invertible, then the unique coefficient vector representing the minimizer is*

$$\xi^\star(u) = G^{-1}b(u), \qquad b_i(u) = \langle u, \phi_i(a) \rangle.$$

*Proof.* The result follows directly from Lemma A.6. Equality is attained if and only if $\hat{u}_\xi(u) = P_{S(a)}u$ almost surely. If $G$ is invertible, the coefficients of $P_{S(a)}u$ in the spanning set $\{\phi_k(a)\}_{k=1}^r$ solve the normal equations $G\xi = b(u)$. $\qquad\square$

To identify the optimal subspace, we use the conditional second-moment operator of the uncentered output field.

**Lemma A.8.** *Under Assumption A.5, define $M_a : \mathcal{U} \to \mathcal{U}$ by*

$$M_a f := \mathbb{E}\big[\langle u, f \rangle u \mid a\big].$$

*Then $M_a$ is self-adjoint, positive semidefinite, and trace-class, with*

$$\mathrm{tr}(M_a) = \mathbb{E}\big[\|u\|^2 \mid a\big].$$

*Therefore $M_a$ admits an orthonormal eigenbasis $(e_k(a))_{k \geq 1}$ with eigenvalues $\lambda_1(a) \geq \lambda_2(a) \geq \cdots \geq 0$.*

*Proof.* For $f, g \in \mathcal{U}$,

$$\langle M_a f, g \rangle = \mathbb{E}\big[\langle u, f \rangle \langle u, g \rangle \mid a\big] = \langle f, M_a g \rangle,$$

so $M_a$ is self-adjoint. Moreover,

$$\langle M_a f, f \rangle = \mathbb{E}\big[\langle u, f \rangle^2 \mid a\big] \geq 0,$$

so $M_a$ is positive semidefinite. For any orthonormal basis $(q_k)_{k \geq 1}$ of $\mathcal{U}$, monotone convergence and Parseval's identity give

$$\sum_{k \geq 1} \langle M_a q_k, q_k \rangle = \mathbb{E}\left[\sum_{k \geq 1} \langle u, q_k \rangle^2 \ \bigg| \ a\right] = \mathbb{E}\big[\|u\|^2 \mid a\big] < \infty.$$

Thus $M_a$ is trace-class, and its trace equals $\mathbb{E}[\|u\|^2 \mid a]$. Since $M_a$ is compact and self-adjoint, the spectral theorem yields the stated eigendecomposition. $\qquad\square$

**Proposition A.9** (Optimal basis: conditional uncentered KL subspace). *Fix $a$. Among all $r$-dimensional subspaces $S \subset \mathcal{U}$,*

$$\inf_{\dim(S)=r} \mathbb{E}\big[\|u - P_S u\|^2 \mid a\big] = \sum_{k > r} \lambda_k(a),$$

*where $(\lambda_k(a), e_k(a))_{k \geq 1}$ are the eigenpairs of $M_a$ from Lemma A.8. The infimum is attained by*

$$S_r^\star(a) = \mathrm{span}\{e_1(a), \ldots, e_r(a)\}.$$

*We refer to this as the conditional uncentered Karhunen–Loève subspace.*

*Proof.* For any $r$-dimensional subspace $S$,

$$\mathbb{E}\big[\|u - P_S u\|^2 \mid a\big] = \mathbb{E}\big[\|u\|^2 \mid a\big] - \mathbb{E}\big[\|P_S u\|^2 \mid a\big].$$

Let $(s_i)_{i=1}^r$ be an orthonormal basis of $S$. Then

$$\mathbb{E}\big[\|P_S u\|^2 \mid a\big] = \sum_{i=1}^r \mathbb{E}\big[\langle u, s_i\rangle^2 \mid a\big]$$

$$= \sum_{i=1}^r \langle M_a s_i, s_i\rangle = \operatorname{tr}(P_S M_a).$$

By the variational characterization of eigenvalues for positive trace-class self-adjoint operators,

$$\sup_{\dim(S)=r} \operatorname{tr}(P_S M_a) = \sum_{k=1}^r \lambda_k(a),$$

with equality attained, for example, by $S = S_r^\star(a) = \operatorname{span}\{e_1(a), \ldots, e_r(a)\}$. Since

$$\operatorname{tr}(M_a) = \sum_{k \geq 1} \lambda_k(a) = \mathbb{E}\big[\|u\|^2 \mid a\big],$$

the minimum residual energy is

$$\sum_{k>r} \lambda_k(a).$$

$\square$

*Remark* A.10 (Uncentered versus centered KL). The subspace above is defined by the second-moment operator $M_a = \mathbb{E}[u \otimes u \mid a]$, not by the centered covariance operator. Hence it is an uncentered KL-type subspace. When the conditional mean is nonzero, the leading directions may capture both mean structure and variability. If the conditional mean vanishes, the construction reduces to the usual centered conditional KL subspace.

We now connect this characterization to the learnable operator encoder by assuming sufficient expressivity of the neural operator basis map and the coefficient encoder at the fixed input $a$.

**Assumption A.11** (Universal approximation for the operator encoder). Let $\mathcal{A}$ and $\mathcal{U}$ be separable Hilbert spaces and fix $r \in \mathbb{N}$. Fix $a \in \mathcal{A}$ and assume $\mathbb{E}[\|u\|^2 \mid a] < \infty$. Consider parametrized maps $\mathtt{NO}_\psi : \mathcal{A} \to \mathcal{U}^r$ and $\mathtt{NF}_\varphi : \mathcal{U} \to \mathbb{R}^r$. Assume:

1. For every $r$-dimensional subspace $S \subset \mathcal{U}$, there exists $\psi$ such that

$$\operatorname{span}(\mathtt{NO}_\psi(a)) = S.$$

2. For every $\Psi \in \mathcal{U}^r$ with $S = \operatorname{span}(\Psi)$, there exists $\varphi$ such that

$$\mathtt{NF}_\varphi(u)^\top \Psi = P_S u, \qquad \mathbb{P}(\cdot \mid a)\text{-a.s.}$$

*Proof of Proposition 4.1.* Fix $\psi$ and define

$$\Psi := \mathtt{NO}_\psi(a) \in \mathcal{U}^r, \qquad S := \operatorname{span}(\Psi).$$

For any $\varphi$, the reconstruction

$$\hat{u}_{\psi,\varphi}(u, a) := \mathtt{NF}_\varphi(u)^\top \Psi$$

lies in $S$. Therefore Proposition A.7 gives

$$\mathcal{L}_{\mathrm{OE}}(\psi, \varphi; a) := \mathbb{E}\big[\|u - \hat{u}_{\psi,\varphi}(u, a)\|^2 \mid a\big] \geq \mathbb{E}\big[\|u - P_S u\|^2 \mid a\big].$$

By Assumption A.11, this lower bound is attainable for the fixed subspace $S$, so

$$\inf_{\varphi} \mathcal{L}_{\mathrm{OE}}(\psi, \varphi; a) = \mathbb{E}\big[\|u - P_{\mathrm{span}(\mathrm{NO}_{\psi}(a))} u\|^2 \mid a\big].$$

Minimizing over $\psi$ and using the universal basis assumption yields

$$\inf_{\psi, \varphi} \mathcal{L}_{\mathrm{OE}}(\psi, \varphi; a) = \inf_{\dim(S)=r} \mathbb{E}\big[\|u - P_S u\|^2 \mid a\big].$$

By Proposition A.9, the optimal subspace is the rank-$r$ conditional uncentered KL subspace spanned by the leading eigenfunctions of $M_a$. This proves the claim. □

## B. Experimental Details

### B.1. Architectural Details

**FNO backbone.** Across all benchmarks, we use FNO as the backbone neural operator. Unless stated otherwise, the backbone maps one input channel to one output channel. In both 1D and 2D, we use hidden width $64$, four Fourier layers, and retain 32 Fourier modes per spatial dimension, namely $[32]$ in 1D and $[32, 32]$ in 2D. The same FNO specification is used to construct the conditioning embedders in DLL.

**FNO dropout.** For the Monte Carlo dropout baseline, we use the same FNO architecture and apply dropout only in the channel MLP blocks, with dropout probability $p = 0.2$. At test time, predictive distributions are approximated by repeated stochastic forward passes. In our experiments, we use $K = 32$ samples.

**PNO.** PNO uses the same FNO backbone, with the final layer outputting two channels for a location scale parameterization. The scale is obtained through a softplus transform. Training uses a reparameterized sampling scheme with an energy score objective. The base configuration uses $8$ samples per training example, while evaluation uses $K = 32$ predictive samples.

**DM and LDM.** For grid based generative baselines, we consider DM and LDM. Both are trained with a flow matching objective and EMA of the model weights. DM uses conditional U-Net backbones in 1D and 2D, with conditioning performed by channel wise concatenation of the conditioning field and the noisy state. LDM uses a two stage pipeline. We first train a VAE style autoencoder with latent width $z_{\mathrm{ch}} = 4$, double latent channel parameterization enabled, and KL weight $10^{-6}$. We then train a conditional U-Net in latent space with the same flow matching objective. In the current configuration, conditioning for LDM is obtained from the frozen autoencoder encoder and concatenated channel wise with the noisy latent.

**DLL.** For DLL, we first train an operator encoder. Its backbone is an FNO that produces an input dependent feature field with $r = 64$ channels, and a learned FNO based output embedder maps target fields to last layer coefficients in $\mathbb{R}^{64}$. We then freeze the operator encoder and train a diffusion model in coefficient space. The velocity model is a conditional MLP with three hidden layers of width $512$, time embedding dimension $32$, and dropout $0.2$. It is conditioned on an FNO embedding of the input. Unless stated otherwise, we evaluate with $K = 32$ samples.

**Conditional U-Net backbone.** For the diffusion based baselines, we use conditional U-Net backbones in 1D and 2D. The 1D model uses base width $32$, and the 2D model uses base width $64$. Both use channel multipliers $(1, 2, 4, 8)$ and two residual blocks at each resolution level. Time is encoded by a sinusoidal embedding of dimension $32$ and injected into every residual block through FiLM style modulation. Attention is applied only at intermediate resolutions, namely resolution 32 in 1D and resolution 16 in 2D. Conditioning is incorporated by concatenating the conditioning features with the noisy input along the channel dimension. When needed, conditioning features are broadcast or resized to match the current spatial resolution. We use SiLU activations and no dropout in the U-Net blocks.

**Parameter counts.** Tables 8 and 9 summarize trainable parameter counts for the backbone models and the additional latent encoders. FNO, FNO dropout, and PNO are matched in parameter budget within each spatial dimension. The diffusion baselines are larger, especially in 2D, whereas DLL is more parameter efficient at the backbone level. Table 9 reports the additional encoder parameters required by LDM and DLL.

*Table 8.* Backbone parameter counts, in millions, for 1D and 2D experiments.

|    | FNO | FNO dropout | PNO | DM | LDM | DLL |
|----|-----|-------------|-----|-----|-----|-----|
| 1D | 0.329M | 0.329M | 0.329M | 3.732M | 3.733M | 2.117M |
| 2D | 8.964M | 8.964M | 8.964M | 34.646M | 34.665M | 10.751M |

*Table 9.* Encoder parameter counts, in millions, for latent representations. AE denotes the autoencoder used by LDM, and OE denotes the operator encoder used by DLL.

|    | LDM (AE) | DLL (OE) |
|----|----------|----------|
| 1D | 3.421M | 0.675M |
| 2D | 9.743M | 17.944M |

## B.2. Training Configurations

**Optimization and normalization.** All models are trained with AdamW using learning rate $10^{-3}$, zero weight decay, cosine learning rate annealing, and gradient clipping with threshold 1.0. We use Gaussian normalization for both inputs and outputs. For stochastic operator learning tasks, input and output normalizers are fitted separately. For DM, LDM, and DLL, we use EMA with decay 0.999.

**Training schedules.** For stochastic operator learning tasks, all methods are trained for 100 epochs. For autoregressive rollout benchmarks, all methods are trained for 500 epochs.

## B.3. Dataset Generation Details

**Stochastic Burgers' equation.** We generate data from the 1D viscous stochastic Burgers' equation on the periodic domain $x \in [0, 2\pi]$,

$$\mathrm{d}u = \left(-\tfrac{1}{2}\,\partial_x(u^2) + \nu\,\partial_{xx}u\right)\mathrm{d}t + \sum_{j \in \{1,3,5\}} w_j \cos(jx)\,\mathrm{d}W_t^j,$$

with viscosity $\nu = 0.1$, and weights $(w_1, w_3, w_5) = (1.0, 0.5, 0.1)$. Space is discretized by a pseudospectral Fourier method on $N = 256$ grid points with two thirds dealiasing. Time integration uses ETDRK4 for the drift and Euler Maruyama for the additive noise. We form one step input output pairs with macro step $\Delta t = 1.0$ and internal step size $\Delta t_{\mathrm{sim}} = 10^{-4}$. Initial conditions are sampled as smooth random Fourier series with coefficients decaying as $k^{-2}$ and are then scaled to unit amplitude. We use 10,000 training inputs and 32 validation and test inputs, with one training output per input and 64 outputs per input for validation and test.

**Stochastic Darcy flow.** We consider Darcy flow on $\Omega = (0,1)^2$ with homogeneous Dirichlet boundary conditions,

$$-\nabla \cdot \big(a(x)\nabla u(x)\big) = f(x), \qquad x \in \Omega,$$
$$u(x) = 0, \qquad x \in \partial\Omega.$$

The domain is discretized on a $128 \times 128$ uniform grid, and the resulting symmetric positive definite linear systems are solved in batches by conjugate gradients with tolerance $10^{-6}$ and maximum iteration count 5000.

The permeability field is sampled as a thresholded Gaussian random field represented in a DCT II basis, which yields a binary field $a(x) \in \{12, 3\}$. Aleatoric uncertainty arises through the random source

$$f(x) = \sigma_{\mathrm{ln}} \exp\big(G_{\mathrm{ln}}(x)\big) + \sigma_{\mathrm{gp}} G_{\mathrm{gp}}(x),$$

where $G_{\mathrm{ln}}$ and $G_{\mathrm{gp}}$ are independent mean zero Gaussian random fields with separable RBF covariances. In our experiments,

$$(\sigma_{\mathrm{ln}}, \ell_{\mathrm{ln}}) = (1.0, 0.2), \qquad (\sigma_{\mathrm{gp}}, \ell_{\mathrm{gp}}) = (9.0, 0.5),$$

with jitter $10^{-5}$ for numerical stability. Dataset sizes match those of stochastic Burgers: 10,000 training inputs and 32 validation and test inputs, with one training output per input and 64 outputs per input for validation and test.

**KS equation.** For the Kuramoto Sivashinsky benchmark, we follow the physical scenario in APEBench (Koehler et al., 2024). Trajectories are generated on a grid of size 256 with output spacing $\Delta t = 1.0$ and 100 internal substeps per saved step. We use 1024 training trajectories of horizon 50 and 128 held out trajectories of horizon 100, each preceded by 100 warmup steps. The held out trajectories are deterministically shuffled and split evenly into validation and test sets.

**Kolmogorov flow.** For the Kolmogorov flow benchmark, we follow the physical scenario in APEBench (Koehler et al., 2024). Trajectories are generated on a $128 \times 128$ grid with output spacing $\Delta t = 0.25$ and 25 internal substeps per saved step. We use 256 training trajectories of horizon 50 and 32 held out trajectories of horizon 100, each preceded by 400 warmup steps. The held out trajectories are deterministically shuffled and split evenly into validation and test sets.

### B.4. Metrics

We report complementary pointwise and distributional metrics for stochastic operator learning, and long horizon forecast metrics for autoregressive rollouts. All fields are flattened when computing sample set distances. When a deterministic prediction is required from a probabilistic model, we use the ensemble mean of $K$ generated samples.

**Stochastic operator learning.** For a fixed conditioning input, let $\{x_k\}_{k=1}^K$ denote predictive samples and $\{y_s\}_{s=1}^S$ denote reference samples. We report the following metrics.

- **Energy distance (ED):**

$$\text{ED}(X,Y) = 2\,\mathbb{E}\|X - Y\|_2 - \mathbb{E}\|X - X'\|_2 - \mathbb{E}\|Y - Y'\|_2,$$

  where $X$, $X'$ are independent draws from the predictive distribution and $Y$, $Y'$ are independent draws from the empirical target distribution.

- **Sliced Wasserstein distance (SWD):**

$$\text{SWD}(X,Y) = \frac{1}{P}\sum_{p=1}^P \mathcal{W}_1\big(\langle X, v_p\rangle, \langle Y, v_p\rangle\big),$$

  where the average is taken over random projection directions $\{v_p\}_{p=1}^P$.

- **NRMSE$_m$ (mean error):** Let $\mu_{\text{pred}} = \frac{1}{K}\sum_{k=1}^K x_k$ and $\mu_{\text{true}} = \frac{1}{S}\sum_{s=1}^S y_s$. We define

$$\text{NRMSE}_m = \frac{\text{RMSE}(\mu_{\text{pred}}, \mu_{\text{true}})}{\sqrt{\mathbb{E}[\mu_{\text{true}}^2]}}.$$

- **NRMSE$_s$ (spread error):** Let $\sigma_{\text{pred}}$ and $\sigma_{\text{true}}$ denote the pointwise ensemble standard deviations of the predictive and reference samples. We define

$$\text{NRMSE}_s = \frac{\text{RMSE}(\sigma_{\text{pred}}, \sigma_{\text{true}})}{\sqrt{\mathbb{E}[\sigma_{\text{true}}^2]}}.$$

**Autoregressive rollouts.** For rollout benchmarks, all metrics are first computed at each forecast step and then averaged over the rollout horizon, excluding the initial condition. For probabilistic models, the deterministic forecast used in pointwise error metrics is the ensemble mean.

- **NRMSE.** At each rollout step, we compute the RMSE between the point prediction and the ground truth field, normalized by the target RMS $L^2$ norm at that step. We then average the normalized error over time.

- **Continuous ranked probability score (CRPS).** For probabilistic forecasts, we report the empirical CRPS,

$$\text{CRPS} = \mathbb{E}|X - y| - \tfrac{1}{2}\,\mathbb{E}|X - X'|,$$

  where $y$ is the ground truth realization and the expectations are approximated using the predicted ensemble.

*Table 10.* **Ablation on coefficient rank for stochastic benchmarks.** Recon denotes the reconstruction NRMSE of the operator encoder. Lower is better for all metrics.

| Dataset | Metric | $r = 16$ | $r = 32$ | $r = 64$ | $r = 128$ |
|---------|--------|----------|----------|----------|-----------|
| Burgers | ED $\downarrow$ | 1.161 | 1.309 | 1.285 | 1.314 |
| | SWD $\downarrow$ | 0.219 | 0.245 | 0.213 | 0.228 |
| | NRMSE$_\text{m}$ $\downarrow$ | 0.260 | 0.279 | 0.252 | 0.238 |
| | NRMSE$_\text{s}$ $\downarrow$ | 0.230 | 0.265 | 0.289 | 0.315 |
| | Recon $\downarrow$ | $6.04\times10^{-2}$ | $5.37\times10^{-2}$ | $4.13\times10^{-2}$ | $3.50\times10^{-2}$ |
| Darcy | ED $\downarrow$ | 0.194 | 0.198 | 0.227 | 0.282 |
| | SWD $\downarrow$ | 0.006 | 0.006 | 0.007 | 0.008 |
| | NRMSE$_\text{m}$ $\downarrow$ | 0.351 | 0.363 | 0.355 | 0.406 |
| | NRMSE$_\text{s}$ $\downarrow$ | 0.314 | 0.318 | 0.357 | 0.503 |
| | Recon $\downarrow$ | $5.48\times10^{-2}$ | $4.19\times10^{-2}$ | $4.11\times10^{-2}$ | $3.72\times10^{-2}$ |

*Table 11.* **Ablation on coefficient rank for deterministic rollout benchmarks.** Lower is better for NRMSE, CRPS, and Recon. SSR is optimal when close to one.

| Dataset | Metric | $r = 16$ | $r = 32$ | $r = 64$ | $r = 128$ |
|---------|--------|----------|----------|----------|-----------|
| KS | NRMSE $\downarrow$ | 0.333 | 0.336 | 0.343 | 0.326 |
| | CRPS $\downarrow$ | 0.460 | 0.458 | 0.470 | 0.448 |
| | SSR $\rightarrow$ 1 | 1.186 | 1.078 | 0.949 | 1.153 |
| | Recon $\downarrow$ | $2.43\times10^{-4}$ | $2.51\times10^{-4}$ | $2.45\times10^{-4}$ | $2.52\times10^{-4}$ |
| Kolmogorov | NRMSE $\downarrow$ | 0.413 | 0.467 | 0.426 | 0.426 |
| | CRPS $\downarrow$ | 0.779 | 0.847 | 0.822 | 0.803 |
| | SSR $\rightarrow$ 1 | 0.846 | 0.732 | 0.620 | 0.765 |
| | Recon $\downarrow$ | $3.41\times10^{-3}$ | $3.41\times10^{-3}$ | $3.25\times10^{-3}$ | $3.51\times10^{-3}$ |

- **Spread skill ratio (SSR).** To assess calibration during rollout, we report

$$\text{SSR} = \frac{\text{Spread}}{\text{RMSE} + \varepsilon}, \qquad \text{Spread} = \sqrt{\mathbb{E}[\text{Var}(X)]}, \qquad \text{RMSE} = \sqrt{\mathbb{E}\big[(\mathbb{E}[X] - y)^2\big]},$$

where $\varepsilon > 0$ is a small constant for numerical stability. Values near 1 indicate that predictive spread is commensurate with forecast error.

## C. Ablation Studies

In this appendix, we provide additional ablation studies for DLL. Unless otherwise specified, we use the same experimental protocol and evaluation metrics as in Appendix B. We study the effect of the coefficient rank $r$, the number of function evaluations (NFE) used during flow matching sampling, and the training dataset size. We also report an additional compatibility experiment using a DeepONet backbone.

### C.1. Ablation on the Coefficient Rank

We first study the effect of the coefficient rank $r$, which determines the dimension of the DLL coefficient space. Tables 10 and 11 report results for $r \in \{16, 32, 64, 128\}$.

On the stochastic benchmarks, increasing $r$ consistently improves the reconstruction error of the operator encoder, indicating that a larger coefficient space provides a more expressive low-rank representation. However, this improvement does not necessarily translate into better distributional fidelity. In particular, ED, SWD, and moment errors are not monotone in $r$. This suggests that after a moderate rank, the additional coefficient dimensions may make the coefficient-space flow-matching model harder to learn without providing a clear gain in the final predictive distribution.

On the deterministic rollout benchmarks, the reconstruction error is already very small and remains largely insensitive to $r$. The rollout metrics also show mixed behavior rather than monotone improvement. These results support the use of a moderate rank, and we use $r = 64$ in the main experiments.

*Table 12.* **Ablation on NFE for stochastic benchmarks.** Lower is better for all metrics.

| Dataset | Metric | NFE = 3 | NFE = 5 | NFE = 10 | NFE = 20 | NFE = 30 | NFE = 50 |
|---|---|---|---|---|---|---|---|
| Burgers | ED ↓ | 2.118 | 1.706 | 1.336 | 1.432 | 1.332 | 1.226 |
| | SWD ↓ | 0.304 | 0.264 | 0.233 | 0.209 | 0.191 | 0.229 |
| | $\text{NRMSE}_m$ ↓ | 0.238 | 0.246 | 0.251 | 0.292 | 0.274 | 0.257 |
| | $\text{NRMSE}_s$ ↓ | 0.481 | 0.394 | 0.300 | 0.281 | 0.270 | 0.243 |
| Darcy | ED ↓ | 0.434 | 0.292 | 0.245 | 0.246 | 0.236 | 0.236 |
| | SWD ↓ | 0.008 | 0.007 | 0.007 | 0.006 | 0.006 | 0.006 |
| | $\text{NRMSE}_m$ ↓ | 0.316 | 0.355 | 0.389 | 0.430 | 0.391 | 0.406 |
| | $\text{NRMSE}_s$ ↓ | 0.506 | 0.402 | 0.374 | 0.379 | 0.421 | 0.419 |

*Table 13.* **Ablation on NFE for deterministic rollout benchmarks.** Lower is better for NRMSE and CRPS. SSR is optimal when close to one.

| Dataset | Metric | NFE = 3 | NFE = 5 | NFE = 10 | NFE = 20 | NFE = 30 | NFE = 50 |
|---|---|---|---|---|---|---|---|
| Kolmogorov | NRMSE ↓ | 0.426 | 0.415 | 0.423 | 0.440 | 0.443 | 0.427 |
| | CRPS ↓ | 0.796 | 0.777 | 0.815 | 0.848 | 0.853 | 0.814 |
| | SSR → 1 | 0.942 | 0.819 | 0.630 | 0.650 | 0.667 | 0.720 |
| KS | NRMSE ↓ | 0.371 | 0.341 | 0.369 | 0.351 | 0.345 | 0.353 |
| | CRPS ↓ | 0.517 | 0.465 | 0.521 | 0.488 | 0.484 | 0.494 |
| | SSR → 1 | 1.751 | 1.328 | 0.865 | 0.798 | 0.777 | 0.681 |

## C.2. Ablation on the Number of Function Evaluations

We next vary the number of function evaluations (NFE) used when solving the learned probability-flow ODE in coefficient space. Tables 12 and 13 report results for $\text{NFE} \in \{3, 5, 10, 20, 30, 50\}$.

On stochastic benchmarks, increasing NFE generally improves distributional fidelity, especially when moving from very small values such as $\text{NFE} = 3$ to moderate values. The improvement saturates after a moderate number of function evaluations, suggesting that long sampling trajectories are not necessary for these PDE benchmarks. On deterministic rollout benchmarks, pointwise accuracy is relatively stable across different values of NFE, while spread-related metrics such as SSR are more sensitive. Overall, DLL achieves acceptable performance with a small or moderate NFE, which helps reduce sampling cost.

## C.3. Ablation on Dataset Size

We also study the effect of training dataset size on the stochastic benchmarks. Table 14 reports results for $N \in \{2500, 5000, 10000\}$.

The dataset size plays an important role in training flow-matching-based generative surrogates. On both stochastic Burgers and stochastic Darcy, increasing the number of training samples generally improves reconstruction and distributional fidelity. In particular, ED, SWD, and reconstruction error decrease as the dataset size grows. These results indicate that DLL benefits from sufficient data when learning conditional predictive distributions. In regimes where data are scarce, incorporating additional physical structure or physics-informed guidance is a promising direction for improving sample efficiency.

## C.4. Compatibility with DeepONet Backbones

Although the main experiments use FNO backbones, DLL is not restricted to FNO. To demonstrate compatibility with another operator architecture, we integrate DLL with a DeepONet backbone on a Darcy inverse problem.

We consider the linear elliptic problem

$$-\nabla \cdot \big(a(x)\nabla u(x)\big) = 1, \qquad x \in \Omega,$$
$$u(x) = 0, \qquad x \in \partial\Omega.$$

In this inverse problem, the conditioning variable is the vector of noisy pressure observations, while the target field is the permeability $a$. We generate 10,000 samples by drawing the permeability field $a$ from a smooth log-Gaussian random field

*Table 14.* **Ablation on dataset size for stochastic benchmarks.** Recon denotes reconstruction NRMSE. Lower is better for all metrics.

| Dataset | Metric | $N = 2500$ | $N = 5000$ | $N = 10000$ |
|---------|--------|------------|------------|-------------|
| Burgers | ED $\downarrow$ | 1.548 | 1.504 | 1.285 |
| | SWD $\downarrow$ | 0.250 | 0.228 | 0.213 |
| | NRMSE$_m$ $\downarrow$ | 0.349 | 0.300 | 0.252 |
| | NRMSE$_s$ $\downarrow$ | 0.273 | 0.294 | 0.289 |
| | Recon $\downarrow$ | $1.137\times10^{-1}$ | $5.713\times10^{-2}$ | $4.132\times10^{-2}$ |
| Darcy | ED $\downarrow$ | 0.367 | 0.241 | 0.227 |
| | SWD $\downarrow$ | 0.009 | 0.007 | 0.007 |
| | NRMSE$_m$ $\downarrow$ | 0.555 | 0.357 | 0.355 |
| | NRMSE$_s$ $\downarrow$ | 0.535 | 0.413 | 0.357 |
| | Recon $\downarrow$ | $9.180\times10^{-2}$ | $6.084\times10^{-2}$ | $4.114\times10^{-2}$ |

*Table 15.* **DLL with a DeepONet backbone.** Results on a Darcy inverse problem. Lower is better for NRMSE and CRPS. SSR is optimal when close to one.

| Method | NRMSE $\downarrow$ | CRPS $\downarrow$ | SSR $\rightarrow$ 1 |
|--------|--------------------|--------------------|---------------------|
| DeepONet | 0.200 | – | – |
| DeepONet-Dropout | **0.194** | 0.142 | 0.232 |
| DeepONet-DLL | 0.197 | **0.121** | **0.783** |

prior on a $64 \times 64$ grid. For each sample, we solve the forward problem and observe noisy point evaluations of $u$ on a fixed $5 \times 5$ sensor grid, using additive Gaussian noise with standard deviation $0.01$. The task is to reconstruct the full permeability field $a$ on the $64 \times 64$ grid from these noisy sensor observations.

When adapting DLL to DeepONet, the output embedder in the operator encoder and the conditioning encoder for the flow-matching model are implemented using simple MLPs. Table 15 shows that DeepONet-DLL achieves reconstruction accuracy comparable to deterministic and dropout baselines, while improving probabilistic metrics such as CRPS and SSR. This result suggests that DLL can be combined with operator backbones beyond FNO.

# D. Qualitative Results

In this section, we visualize conditional samples of the target field generated by our model across our experimental benchmarks.

## D.1. Stochastic Problems

Figure 3 qualitatively compares conditional predictions on the stochastic Burgers' equation. The ground truth column exhibits substantial sample-to-sample variability, which is also reflected in the nontrivial standard-deviation band. In contrast, the FNO family produces predictions that are either nearly deterministic or exhibit weakly structured dispersion: although the mean trend can be reasonable, the sampled trajectories concentrate around it and the resulting uncertainty bands are largely uninformative, indicating a failure to capture the conditional spread of the target distribution. Probabilistic baselines such as diffusion in pixel space and its latent variants generate diverse samples, but their variability is less consistently aligned with the ground truth heteroscedastic structure. Our DLL, by operating in an operator-encoder coefficient space, produces samples whose fluctuations follow the correct spatial dependence and yields uncertainty bands that closely match the ground truth, demonstrating substantially improved modeling of meaningful predictive spread.

Figure 4 presents a qualitative comparison on the stochastic Darcy flow benchmark, reporting the predictive mean (top row) and the predictive standard deviation (bottom row) for a representative test case. The ground truth exhibits spatially structured variability, indicating that uncertainty is strongly heterogeneous across the domain. In contrast, the FNO-based approaches, including deterministic FNO, dropout-augmented FNO, and PNO, do not recover a meaningful uncertainty structure. In our experiments these baselines also tend to overfit, and model selection based on the best validation checkpoint often returns early, non-converged states whose mean and variance maps are qualitatively inconsistent with the target statistics. By comparison, DLL produces both a mean field that matches the large-scale solution structure and a standard-deviation map that aligns with the ground truth spatial pattern. This indicates that DLL successfully learns distributional information

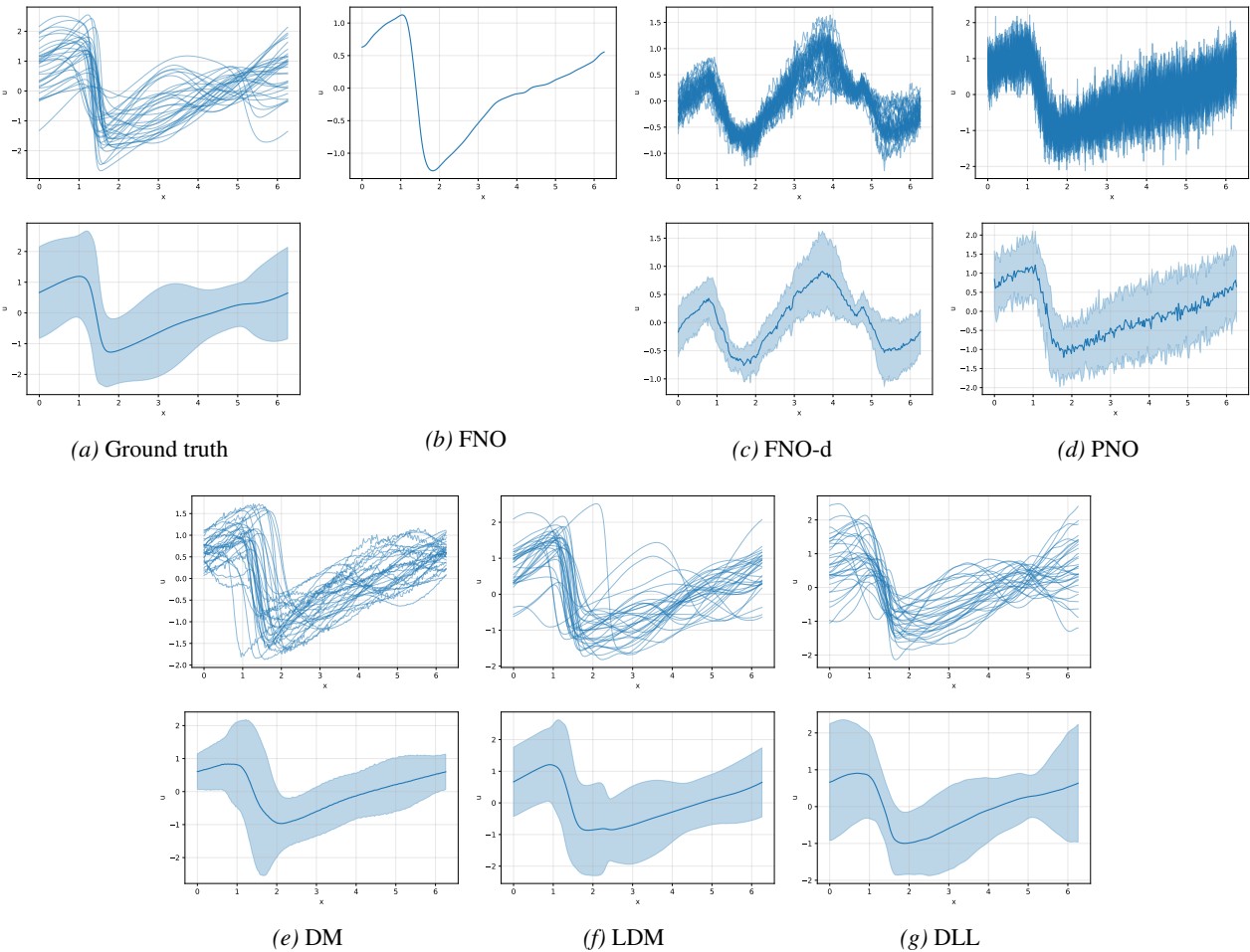

*Figure 3.* Stochastic Burgers' equation. Columns compare the ground truth and different surrogate models. For each method, the top panel shows multiple realizations of the solution field $u(x)$ for a fixed input, illustrating sample diversity. The bottom panel shows the predictive mean (solid line) and an uncertainty band given by standard deviation (shaded region) estimated from the samples.

through explicit conditional generative modeling in the operator-encoder coefficient space, rather than relying on implicit or weak stochasticity in the backbone.

## D.2. Autoregressive Rollouts

Figure 5 compares long horizon rollout predictions for the KS system at the 50th step, overlaying the ground truth (black) with each method's predictive mean (blue) and a standard deviation band computed from generated samples (shaded). While FNO and PNO remain reasonably accurate in terms of the mean trajectory, their uncertainty bands are systematically too narrow relative to the observed mismatch to the ground truth, indicating overconfident uncertainty estimates under chaotic error amplification. Diffusion-based baselines yield wider spreads but may sacrifice mean fidelity. In contrast, DLL preserves competitive accuracy while producing a visibly more informative spread around the mean, better reflecting rollout uncertainty and providing more credible uncertainty quantification in this long horizon regime.

Figure 6 reports qualitative rollout results for Kolmogorov flow at the 50th prediction step, showing the predictive mean (top), predictive standard deviation (middle), and the pointwise error for the same test case (bottom). A key observation is that diffusion-based generative surrogates produce uncertainty maps that meaningfully track where the rollout is difficult: regions with larger pointwise error tend to coincide with elevated predictive standard deviation. This correlation is visible for both pixel space and latent diffusion baselines and is especially clear for DLL, whose uncertainty highlights the same coherent structures that dominate the error field while maintaining a competitive mean prediction. In contrast, non-generative baselines often yield weakly structured or poorly aligned uncertainty patterns, suggesting that explicit generative modeling

is important for producing uncertainty estimates that reflect rollout error under long horizon dynamics.

Figure 7 summarizes long horizon rollout performance on KS (top) and Kolmogorov flow (bottom) using complementary accuracy and uncertainty metrics. NRMSE (left) and CRPS (middle) generally increase with rollout step, reflecting error accumulation and growing distributional mismatch in chaotic dynamics.

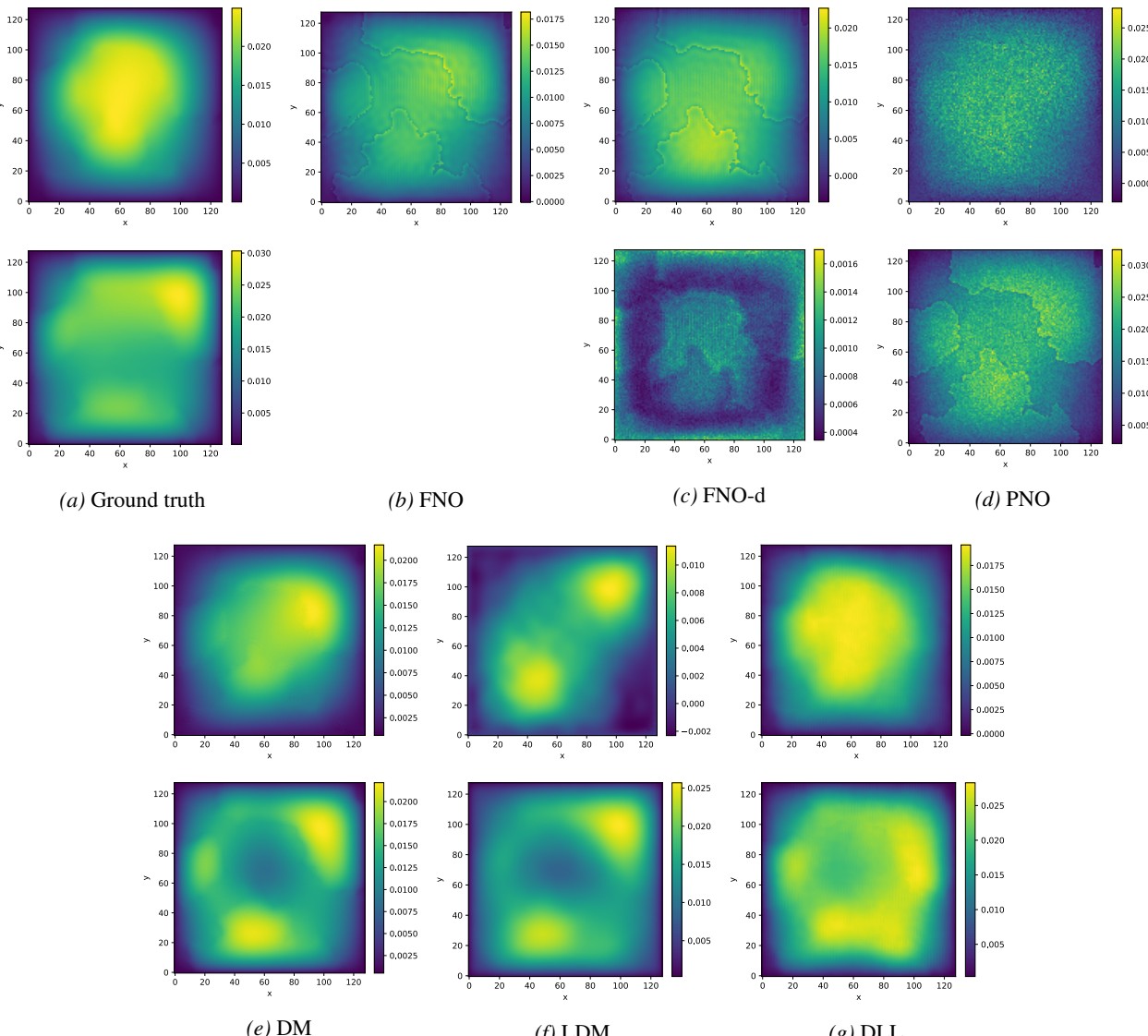

*Figure 4.* Stochastic Darcy flow. For each method, we generate conditional samples of the solution field and summarize them by the sample mean (top) and per-pixel sample standard deviation (bottom). This visualization highlights both accuracy of the central prediction and the spatial structure of predictive uncertainty.

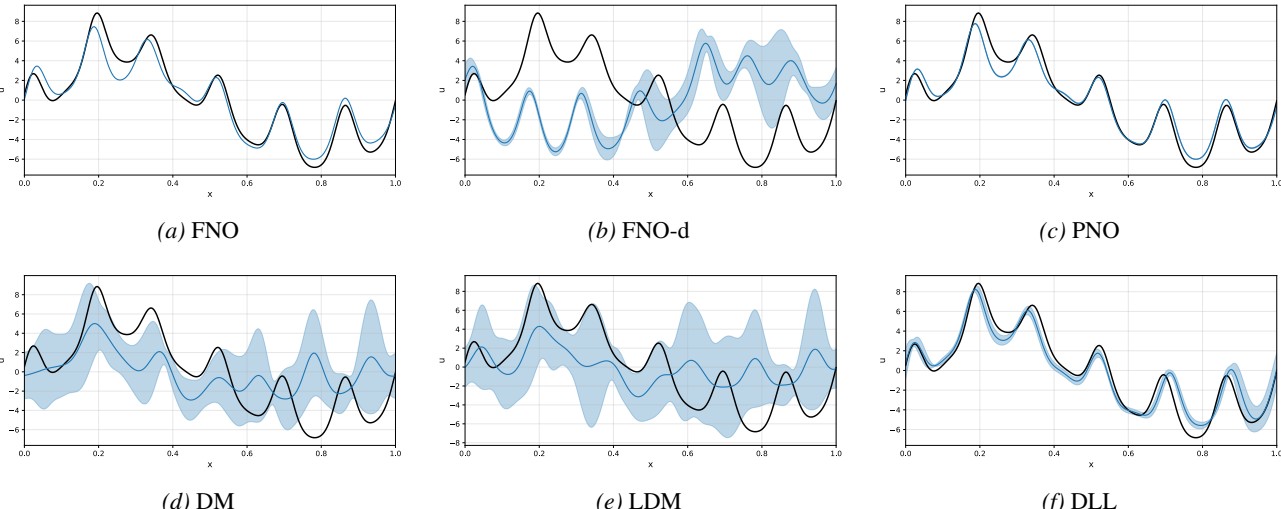

*Figure 5.* KS equation. Long horizon rollout comparison at rollout step 50. The black curve denotes the ground truth solution, while the blue curve shows the predictive mean of each method. Shaded regions indicate predictive standard deviation estimated from generated samples.

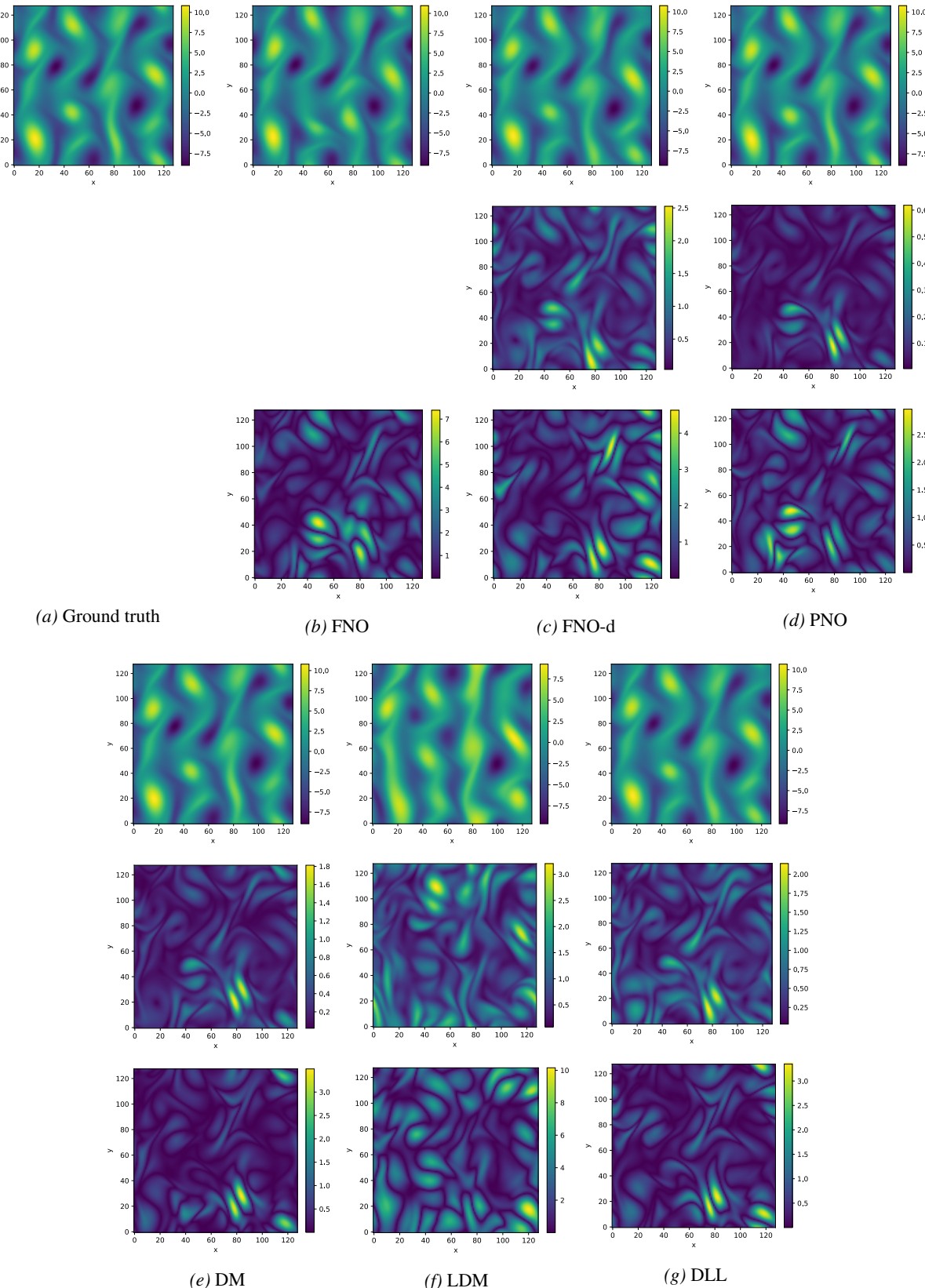

*Figure 6.* Kolmogorov flow. Rollout evaluation at the 50th step. Rows show the predictive mean (top), predictive standard deviation (middle), and the pointwise absolute error (bottom) for a representative test case across baselines and DLL.

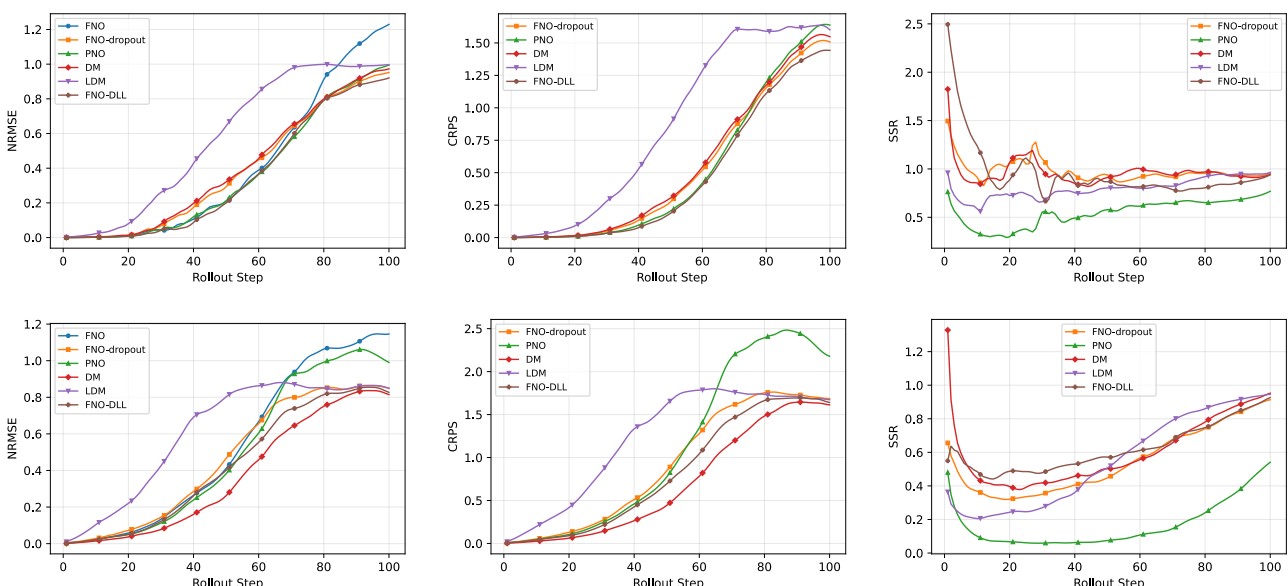

*Figure 7.* Rollout evaluation on KS (top row) and Kolmogorov flow (bottom row). We report NRMSE (left), CRPS (middle), and SSR (right) as a function of rollout step for baselines and DLL, illustrating long horizon accuracy and predictive spread.

