# OpenReview forum: "Generative Neural Operators through Diffusion Last Layer"
_ICML.cc/2026/Conference — ICML 2026 regular_

### Official Review · Reviewer_CBGL · 2026-03-10

**Soundness:** 2
**Presentation:** 3
**Significance:** 3
**Originality:** 2
**Overall Recommendation:** 4
**Confidence:** 4

**Summary:**

The paper proposes Diffusion Last Layer (DLL), a lightweight probabilistic head that can be attached to an arbitrary neural operator backbone to turn a deterministic operator into a conditional generative surrogate. The core idea is to first learn an operator encoder that represents outputs with an input-dependent low-rank basis and coefficient vector, and then run diffusion in the low-dimensional coefficient space rather than directly in pixel/function space. Empirically, the method is evaluated on stochastic Burgers, stochastic Darcy, Kuramoto–Sivashinsky, and Kolmogorov flow, where DLL is reported to improve distributional fidelity and often offer a favorable trade-off between accuracy, calibration, and rollout stability.

**Compliance With Llm Reviewing Policy:**

Affirmed.

**Final Justification:**

The authors’ response has addressed most of my concerns, and I have increased my score to 4.

**Key Questions For Authors:**

1. I am not entirely clear about the computational cost of training the encoder components (e.g., NO and NF). In addition, it would be helpful to understand whether the learned representation truly corresponds to the KL decomposition basis. For instance, are the learned basis functions orthogonal, as in the classical KL expansion? More generally, what structural properties do the learned “basis functions” possess?

2. I would also appreciate it if the authors could address the third point raised in the "Weaknesses" section regarding the efficiency of generative approaches for forward problems in scientific computing.

3. Discussion: the proposed low-rank decomposition structure appears somewhat similar to the architecture used in DeepONet. Could the authors clarify the main differences between their approach and DeepONet, particularly in terms of the learned representation and model design?

4. Another minor question that confuses me concerns the terminology “diffusion last layer.” The idea of encoding the generation target into a latent space and then performing generation in that latent space has been studied for quite some time, and the paper also discusses related approaches. My understanding is that the proposed method follows a similar paradigm. In that case, why is the method referred to as a diffusion last layer? In particular, where does the notion of the “last layer” mainly manifest in the model architecture or training procedure?

**Limitations:**

yes

**Strengths And Weaknesses:**

## Strengths
1. The paper proposes a new encoding–decoding approach. Overall, the key idea is to perform a dimensionality reduction of the output function. An interesting aspect is that the method replaces the commonly predefined basis functions with learnable basis functions, which depend only on the input a. The main contribution lies in introducing an effective encoding mechanism under this formulation.

2. Generative methods have unique advantages in scientific computing, such as in sparse reconstruction, inverse problems, and uncertainty quantification, and the field has been developing rapidly. This paper further extends the application of such generative approaches in scientific computing.

## Weakness

1. The paper establishes a theoretical connection between the operator encoder and the truncated conditional KL subspace. However, it does not empirically verify whether the learned basis and coefficient representations actually align with the conditional KL decomposition in practice.

2. For the stochastic experiments, such as the stochastic Burgers equation, the paper mainly evaluates the consistency of the predicted distributions. However, for the forward problem of stochastic PDEs, a more meaningful task is typically to predict the solution \(u_t\) (or the final state) given a realization of \(W_t\) together with the initial condition. Directly predicting the distribution at time \(T\) does not seem to have strong practical relevance in this context. In contrast, the experimental setup for the stochastic Darcy flow problem appears more appropriate.

3. Most of the experiments focus on forward problems of PDE/SPDEs. However, the computational efficiency of using generative approaches for solving forward problems in scientific computing is not discussed. In particular, when autoregressive rollout is used, generation-based approaches may be relatively inefficient, especially for deterministic problems. In contrast, generative methods often demonstrate clearer advantages in inverse problems. It may therefore be worthwhile for the authors to reconsider the problem formulation and experimental design.

---

> ### Author Rebuttal · Authors · 2026-03-30
>
> We appreciate the reviewer's valuable comments.
>
> **Q1 and W1: Training cost and connection with KL expansion of $\mathtt{OE}$**
>
> The training cost of $\mathtt{OE}$ is comparable to training a regression model with the same $\mathtt{NO}$ and $\mathtt{NF}$ architectures. While training the encoder is more expensive than training the diffusion model in the latent space, it significantly reduces the overall cost by enabling diffusion in a low-dimensional space with a lightweight MLP denoiser. As in latent diffusion, this leads to improved efficiency compared to pixel-space diffusion. For example, in the Kolmogorov flow experiment, pixel-space diffusion requires roughly one day on a single A6000, while DLL and latent diffusion (including encoder training) take about half a day.
>
> Regarding the KL connection, our claim is not that the learned basis functions are strictly orthogonal or identical to the classical KL eigenfunctions. Rather, the theorem assumes full access to the data distribution and shows that, in this ideal population setting, the optimal input-dependent rank-$r$ subspace is the truncated KL subspace of the conditional solution distribution. In practice, with only finite samples and no explicit orthogonality constraint, the learned basis functions need not be orthogonal. Thus, our claim is about recovering an approximately optimal low-rank subspace, not the exact classical KL basis.
>
>
> **Q2 and W3: Inefficiency in using generative models for forward problems**
>
> We agree that generative approaches introduce additional computational overhead compared to deterministic models. However, this overhead is not prohibitive, and such models still offer advantages even in deterministic settings, including improved rollout stability and uncertainty quantification. Accordingly, there has been growing interest in diffusion-based surrogate simulators for long-horizon forward prediction (e.g., PDE-Refiner[1], GenCast[2]).
>
> In PDE settings, the required number of function evaluations (NFE) is typically small, as also shown in the response to the  **Reviewer rSeH, Q4**. Moreover, DLL improves efficiency by performing diffusion in a low-dimensional latent space with a lightweight MLP denoiser, reducing per-step cost compared to pixel-space diffusion. Thus, the computational overhead is moderate, while retaining the benefits of generative modeling.
>
> **Q3: Comparison with DeepONet**
>
> While DLL and DeepONet share a superficially similar low-rank structure, their roles and formulations are fundamentally different. DeepONet represents a deterministic operator by combining branch (coefficients from input sensors) and trunk (basis functions over spatial coordinates) networks. In contrast, DLL learns an input-dependent functional basis (via the operator encoder) and models random coefficients through a conditional diffusion process, enabling a distribution over outputs rather than a single deterministic prediction.
>
> Thus, DLL can be viewed as extending this low-rank structure to a probabilistic setting, where both the basis functions and coefficient distribution are conditioned on the input.
>
> Notably, DLL is modular and can be combined with DeepONet as a backbone, as demonstrated in our response to **Reviewer nCAD, Q2**.
>
>
> **Q4: Clarification on the term "diffusion last layer"**
>
> The term diffusion last layer emphasizes a modular perspective. Diffusion is used as a probabilistic output head within a neural-operator architecture, rather than as a standalone generative model.
>
> The backbone first produces an input dependent functional representation via the operator encoder. Diffusion is then applied only to the final coefficient representation, replacing a deterministic output layer with a stochastic one. Thus, last layer refers to diffusion operating on the final latent coefficients, both architecturally and in the training pipeline.
>
>
> **W2: Stochastic Burgers equation setting is impractical**
>
> We agree that predicting trajectories conditioned on $W_t$ is meaningful. Our focus, however, is to evaluate uncertainty modeling when the stochastic forcing is unobserved, where predicting the distribution at time $T$ provides a natural measure of aleatoric uncertainty and distributional fidelity.
>
> [1] P. Lippe, et al. "Pde-refiner: Achieving accurate long rollouts with neural pde solvers." Neurips, 2023.
> \
> [2] I. Price, et al. "Probabilistic weather forecasting with machine learning." Nature, 2025.

---

### Official Review · Reviewer_nCAD · 2026-03-12

**Soundness:** 4
**Presentation:** 3
**Significance:** 4
**Originality:** 3
**Overall Recommendation:** 5
**Confidence:** 4

**Summary:**

The paper proposes a method called Diffusion Last Layer (DLL) to model conditional distributions for operator learning tasks, which is crucial for Uncertainty Quantification (UQ) in stochastic systems. DLL works as an add-on to existing neural operator architectures, adding latent diffusion (more specifically Flow Matching) to a low-rank latent representation of the target output. The method entails training an input-dependent encoder of the output space, and using this latent representation to carry out diffusion conditioned on the input function $a$ to produce samples of the target output distribution. The authors validate their method on both one-shot and autoregressive operator learning tasks, where they find good performance when compared with other methods.

**Compliance With Llm Reviewing Policy:**

Affirmed.

**Final Justification:**

This paper introduces novel and interesting ideas for implementing diffusion in operator learning problems and recommend acceptance.

**Key Questions For Authors:**

1. How are the data requirements for the method? Diffusion tends to be data-hungry, which may limit the applicability to scientific workloads where data generation is expensive. I see in line 326 that you use 10,000 input/output pairs per system. It would be interesting to consider an ablation on the dataset size and model performance for smaller train sizes (potentially in the appendix).
2. Most operator learning architectures are either inspired by the FNO approach and the Neural Fields approach (like DeepONets, etc). This work primarily focuses on the former. Do you have any thoughts on how compatible this is with architectures based on neural fields? What is the same and what changes?
3. [line 103] Do you have a formal definition for a “space of random functions”? I am not sure I exactly understand this mathematical object. Is that meant to be the space of distributions over $\mathcal{U}$?

**Limitations:**

I think the paper generally neglects discussion of limitation and negative societal impact. It might be interesting to contextualize data and compute requirements, and potential negative impacts including weapons/military applications.

**Strengths And Weaknesses:**

Strengths
* The methodology proposed is well-motivated and helps address the issue of aleatoric UQ in operator learning, which is important but often neglected in the literature.
* The DLL approach is flexible and can be deployed across a wide range of operator learning approaches and architectures.
* The paper is well-written and presents its main ideas in a clear and approachable way.
* The paper evaluates experiments both on one-shot PDE problems (Burgers + Darcy) and more challenging autoregressive systems (KS + Kolmogorov Flow).

Weaknesses
* Despite including both one-shot and autoregressive tasks, Burgers and Darcy problems are fairly simple benchmarks, and there is room for including more complex tasks.
* Due to the data-hungry nature of diffusion models and the limited data availability in scientific domains, the method may not be applicable to scenarios where data is scarce.

---

> ### Author Rebuttal · Authors · 2026-03-30
>
> We appreciate the reviewer's valuable comments.
>
> **Q1: Ablations on dataset size**
>
> Dataset size plays a critical role in training diffusion models. On both stochastic benchmarks, increasing the dataset size consistently improves reconstruction and distributional fidelity (lower ED/SWD and Recon NRMSE), highlighting the importance of sufficient data for learning accurate generative surrogates. While performance improves with data, in settings where strong physical priors are available, incorporating physics-informed guidance at inference time may further enhance generalization.
>
> | Dataset | Metric | N=2500 | N=5000 | N=10000 |
> |-|-|-|-|-|
> | Burgers | ED | 1.548 | 1.504 | 1.285 |
> |  | SWD | 0.250 | 0.228 | 0.213 |
> |  | NRMSE (mean) | 0.349 | 0.300 | 0.252 |
> |  | NRMSE (std) | 0.273 | 0.294 | 0.289 |
> |  | Recon | $1.137\times10^{-1}$ | $5.713\times10^{-2}$ | $4.132\times10^{-2}$ |
> | Darcy | ED | 0.367 | 0.241 | 0.227 |
> |  | SWD | 0.009 | 0.007 | 0.007 |
> |  | NRMSE (mean) | 0.555 | 0.357 | 0.355 |
> |  | NRMSE (std) | 0.535 | 0.413 | 0.357 |
> |  | Recon | $9.180\times10^{-2}$ | $6.084\times10^{-2}$ | $4.114\times10^{-2}$ |
>
>
> **Q2: Compatibility with neural fields backbones**
>
> These results suggest that DLL can be extended beyond FNO to other function-output architectures, such as DeepONet. To demonstrate this, we integrate DLL with a DeepONet backbone on a Bayesian inverse problem for Darcy flow. We consider the linear elliptic problem
> $$-\nabla\cdot\big(a(x)\nabla u(x)\big)=1, \qquad x\in\Omega,$$
> with homogeneous Dirichlet boundary conditions and generate 10,000 samples with $a$ drawn from a smooth log-Gaussian random field prior on a $64\times64$ grid. For each sample, we solve the forward problem and observe noisy point evaluations of $u$ on a fixed $5\times5$ sensor grid (additive Gaussian noise with standard deviation $0.01$). DeepONet then maps these low-resolution observations to a full-resolution reconstruction of the permeability field $a$ at $64\times64$ resolution.
>
> A key design choice when adapting DLL to different backbones is the selection of (i) the output embedder for the operator encoder and (ii) the input encoder for the diffusion model. In this DeepONet setting, we use simple MLPs for both.
>
> Compared with deterministic and dropout baselines, DLL achieves comparable reconstruction accuracy while improving predictive uncertainty metrics on this inverse problem.
>
> **DLL with DeepONet backbone**
> ||NRMSE|CRPS|SSR|
> |-|-|-|-|
> |DeepONet|0.200|-|-|
> |DeepONet-Dropout|0.194|0.142|0.232|
> |DeepONet-DLL|0.197|0.121|0.783|
>
>
> **Q3: Definition of space of random function**
>
> We thank the reviewer for pointing out this ambiguity. We will revise Section 2.1 to make clear that $\mathcal{G}^\ddagger$ is intended as a deterministic map from inputs to probability measures on $\mathcal{U}$, i.e.,
> $$
> \mathcal{G}^\ddagger: \mathcal{A} \rightarrow \mathcal{P}(\mathcal{U})
> $$
> rather than to individual $\mathcal{U}$-valued random variables. Thus, $\mathcal{G}^\ddagger (a)$ denotes the conditional law of the output function given $a$. This clarification is consistent with the current text in Section 2.1, which already describes the stochastic operator as a conditional distribution and the deterministic case as a Dirac measure.

---

> > ### Author Rebuttal · Reviewer_nCAD · 2026-04-03
> >
> > Thank you to the authors for taking the time to respond. I believe the paper is in a good state and maintain my positive recommendation to accept it to the conference.

---

### Official Review · Reviewer_rSeH · 2026-03-13

**Soundness:** 3
**Presentation:** 2
**Significance:** 3
**Originality:** 2
**Overall Recommendation:** 4
**Confidence:** 3

**Summary:**

This paper introduces the Diffusion Last Layer (DLL), a lightweight probabilistic head designed to be attached to arbitrary neural operator (NO) backbones to enable uncertainty quantification (UQ) and generative modeling. The core innovation lies in the use of an operator encoder that represents the target function space through a low-rank Karhunen-Loève expansion. Specifically, the NO backbone generates input-dependent basis functions, while a conditional diffusion model (trained in a compact coefficient space) models the stochasticity of the output. The authors provide theoretical justification linking the velocity-matching objective of diffusion to the Wasserstein distance and demonstrate that the operator encoder recovers an optimal rank-$r$ reconstruction. Experiments on stochastic and deterministic benchmarks show that DLL outperforms existing deterministic and probabilistic baselines.

**Compliance With Llm Reviewing Policy:**

Affirmed.

**Final Justification:**

My concerns have been adequately addressed, so I raise my rating to 4.

**Key Questions For Authors:**

1. The study uses a fixed rank $r=64$ for all experiments. How sensitive is the model's performance to this hyperparameter? Could you provide an ablation study on the trade-off between rank size, computational cost, and the capture of high-frequency stochastic features?

2. The paper claims DLL is architecture-agnostic, yet results are only reported for FNO. Have you tested DLL with other backbones like DeepONet or Transformer-based operators?

3. The authors utilize a two-stage training process (freezing the operator encoder before training the diffusion head). Did you experiment with joint training, and how critical is the quality of the initial frozen encoder to the final generative performance?

4. While the diffusion head is described as lightweight, both training and inference with diffusion models typically require multiple iterative steps to achieve strong performance. However, the authors only use $T=10$ steps. It would be helpful for the authors to discuss the trade-off between computational overhead and performance when varying $T$.

**Limitations:**

yes

**Strengths And Weaknesses:**

Strengths:
- The integration of Karhunen-Loève expansion principles with neural operators is well-motivated. By performing diffusion in a latent coefficient space defined by the operator itself, the model preserves discretization-invariance—a key property of neural operators that is often lost in pixel-space diffusion.
- DLL can be attached to arbitrary neural operator (NO) backbones to enable uncertainty quantification (UQ) and generative modeling.
- DLL achieves strong empirical results on both stochastic and deterministic benchmarks, outperforming existing deterministic and probabilistic baselines.

Weaknesses:
- The performance of the model is inherently tied to the rank $r$ of the Karhunen-Loève expansion. While the authors provide theoretical justification for the optimality of the encoder's reconstruction for a given rank, the empirical evaluation is restricted to a fixed value of $r=64$ across all benchmarks. The paper would be significantly strengthened by an ablation study demonstrating how varying $r$ influences the trade-off between computational efficiency and the model's ability to capture fine-grained stochastic features.
- The authors claim that the Diffusion Last Layer is architecture-agnostic and can be integrated with arbitrary neural operator (NO) backbones, such as DeepONet. However, the experimental evaluation is exclusively conducted using a FNO backbone. Providing results with at least one alternative backbone would better support the claim of the method's generalizability and modular nature.

---

> ### Author Rebuttal · Authors · 2026-03-30
>
> We appreciate the reviewer's valuable comments.
>
> **Q1: Ablations on varying $r$**
>
> We conduct an ablation study over the latent rank $r$. On stochastic benchmarks, increasing $r$ consistently improves reconstruction (lower Recon NRMSE), indicating a more expressive operator encoder. However, this improvement does not translate into better distributional fidelity.
>
> In contrast, for deterministic tasks, reconstruction error is already negligible and largely insensitive to $r$, since the target is effectively a point estimate and can be captured with a low-rank representation. For the rollout performance, the effect of $r$ is mixed rather than monotone.
>
> These results suggest that increasing $r$ beyond a moderate range does not provide a consistent benefit and can make diffusion in coefficient space harder to learn.
>
> **Stochastic benchmarks**
>
> | Dataset | Metric | r=16 | r=32 | r=64 | r=128 |
> |-|-|-|-|-|-|
> | Burgers | ED | 1.161 | 1.309 | 1.285 | 1.314 |
> |  | SWD | 0.219 | 0.245 | 0.213 | 0.228 |
> |  | NRMSE (mean) | 0.260 | 0.279 | 0.252 | 0.238 |
> |  | NRMSE (std) | 0.230 | 0.265 | 0.289 | 0.315 |
> |  | Recon | $6.04\times10^{-2}$ | $5.37\times10^{-2}$ | $4.13\times10^{-2}$ | $3.50\times10^{-2}$ |
> | Darcy | ED | 0.194 | 0.198 | 0.227 | 0.282 |
> |  | SWD | 0.006 | 0.006 | 0.007 | 0.008 |
> |  | NRMSE (mean) | 0.351 | 0.363 | 0.355 | 0.406 |
> |  | NRMSE (std) | 0.314 | 0.318 | 0.357 | 0.503 |
> |  | Recon | $5.48\times10^{-2}$ | $4.19\times10^{-2}$ | $4.11\times10^{-2}$ | $3.72\times10^{-2}$ |
>
> **Deterministic rollout benchmarks**
>
> | Dataset | Metric | r=16 | r=32 | r=64 | r=128 |
> |-|-|-|-|-|-|
> | KS | NRMSE | 0.333 | 0.336 | 0.343 | 0.326 |
> |  | CRPS | 0.460 | 0.458 | 0.470 | 0.448 |
> |  | SSR | 1.186 | 1.078 | 0.949 | 1.153 |
> |  | Recon | $2.43\times10^{-4}$ | $2.51\times10^{-4}$ | $2.45\times10^{-4}$ | $2.52\times10^{-4}$ |
> | Kolmogorov Flow | NRMSE | 0.413 | 0.467 | 0.426 | 0.426 |
> |  | CRPS | 0.779 | 0.847 | 0.822 | 0.803 |
> |  | SSR | 0.846 | 0.732 | 0.620 | 0.765 |
> |  | Recon | $3.41\times10^{-3}$ | $3.41\times10^{-3}$ | $3.25\times10^{-3}$ | $3.51\times10^{-3}$ |
>
>
> **Q2: Compatibility with other backbones**
>
> We additionally conduct experiments with a DeepONet backbone and refer to the response to **Reviewer nCAD, Q2** for details.
>
> **Q3: Why two stage?**
>
> Diffusion training requires samples from a well-defined target distribution. In our setting, this corresponds to the latent coefficients, which are only meaningful after the operator encoder is trained. Thus, joint training is impractical. Following standard latent diffusion, we first train the encoder, then train diffusion on the fixed latent space to ensure a stable target distribution.
>
> **Q4: Ablations on varying $T$**
>
> For PDE tasks, target distributions are typically low-dimensional and less multimodal than images, enabling stable performance even with small $T$. On stochastic benchmarks, increasing $T$ slightly improves distributional fidelity, though gains saturate beyond moderate $T$. For deterministic rollout tasks, point accuracy is relatively stable across $T$, whereas spread-related metrics more sensitive. Overall, acceptable performance is already achieved with relatively small $T$, suggesting that very long diffusion trajectories are not necessary.
>
> **Stochastic benchmarks**
>
> | Dataset | Metric | T=3 | T=5 | T=10 | T=20 | T=30 | T=50 |
> |-|-|-|-|-|-|-|-|
> | Burgers | ED | 2.118 | 1.706 | 1.285 | 1.432 | 1.332 | 1.226 |
> |  | SWD | 0.304 | 0.264 | 0.213 | 0.209 | 0.191 | 0.229 |
> |  | NRMSE (mean) | 0.238 | 0.246 | 0.252 | 0.292 | 0.274 | 0.257 |
> |  | NRMSE (std) | 0.481 | 0.394 | 0.289 | 0.281 | 0.270 | 0.243 |
> | Darcy | ED | 0.434 | 0.292 | 0.227 | 0.246 | 0.236 | 0.236 |
> |  | SWD | 0.008 | 0.007 | 0.007 | 0.006 | 0.006 | 0.006 |
> |  | NRMSE (mean) | 0.316 | 0.355 | 0.355 | 0.430 | 0.391 | 0.406 |
> |  | NRMSE (std) | 0.506 | 0.402 | 0.357 | 0.379 | 0.421 | 0.419 |
>
> **Deterministic rollout benchmarks**
>
> | Dataset | Metric | T=3 | T=5 | T=10 | T=20 | T=30 | T=50 |
> |-|-|-|-|-|-|-|-|
> | KS | NRMSE | 0.371 | 0.341 | 0.343 | 0.351 | 0.345 | 0.353 |
> |  | CRPS | 0.517 | 0.465 | 0.470 | 0.488 | 0.484 | 0.494 |
> |  | SSR | 1.751 | 1.328 | 0.949 | 0.798 | 0.777 | 0.681 |
> | Kolmogorov Flow | NRMSE | 0.426 | 0.415 | 0.426 | 0.440 | 0.443 | 0.427 |
> |  | CRPS | 0.796 | 0.777 | 0.822 | 0.848 | 0.853 | 0.814 |
> |  | SSR | 0.942 | 0.819 | 0.620 | 0.650 | 0.667 | 0.720 |

---

> > ### Author Rebuttal · Reviewer_rSeH · 2026-04-04
> >
> > Thank you for your detailed rebuttal. My concerns have been sufficiently resolved, so I will raise the score to 4.

---

### Official Review · Reviewer_dXc6 · 2026-03-20

**Soundness:** 3
**Presentation:** 3
**Significance:** 3
**Originality:** 3
**Overall Recommendation:** 4
**Confidence:** 4

**Summary:**

The paper introduces the *Diffusion Last Layer* (DLL), a two-stage probabilistic extension for neural operators: a backbone NO is retrained to produce an input-dependent basis $\Phi(a)$, and a lightweight MLP diffusion head is trained in the resulting low-dimensional Karhunen–Loève (KL) coefficient space to sample $p(u \mid a)$. The method is supported by a Wasserstein stability bound and a KL-subspace optimality result, and evaluated on four PDE benchmarks against deterministic, Bayesian, and generative operator baselines.

**Compliance With Llm Reviewing Policy:**

Affirmed.

**Final Justification:**

- DLL attaches a lightweight diffusion head in the basis coefficient space of a neural operator encoder to model conditional output distributions. The core idea is clean, and backbone-agnostic.

- The authors agreed to revise the overclaimed plug-and-play framing, the unsubstantiated epistemic-uncertainty claim, and the calibration language. My main residual concern is baseline coverage. The authors acknowledge CoNFiLD, WDNO, and neural processes are relevant but cite adaptation costs. I find this inconsistent with including DM and LDM, which also required adaptation. Without at least one additional conditional generative baseline in a related latent/spectral space, the empirical positioning remains incomplete.

- Note. I did not verify the proofs and assumptions in detail; my evaluation focuses on the methodological design, empirical positioning, and framing.

- Bottom line. Sound contribution. Competitive but undercontextualised empirical results due to the narrow baseline set. The rebuttal addressed framing concerns but did not close the empirical positioning gap.

**Key Questions For Authors:**

1. **Theory--experiment alignment.**
Proposition 1 bounds $\mathcal{W}_2$, yet experiments report SWD (sliced $\mathcal{W}_1$) and ED. Could the authors additionally report sliced $\mathcal{W}_2$ for consistency?

2. **Reconstruction bottleneck.**
The operator encoder reconstructs Burgers targets with NRMSE $\approx 4\times10^{-2}$, roughly $20\times$ worse than the standard autoencoder at higher compression ($4\times$ vs. $2\times$). Does replacing `OE` with `AE` inside the full DLL pipeline improve generation quality on Burgers? This ablation would clarify whether the reconstruction gap is a genuine bottleneck.

3. **Sensitivity to latent dimension $r$.**
How does generation quality (ED, SWD, NRMSE) change as $r$ varies (e.g., $r\in\{16,32,64,128\}$)?

4. **Underperformance on Kolmogorov flow.**
Pixel-space `DM` outperforms `DLL` on Kolmogorov flow in both NRMSE and CRPS. What do the authors attribute this to?

5. **Calibration on stochastic benchmarks.**
The introduction claims "better-calibrated uncertainty" on stochastic benchmarks, but only distributional fidelity metrics (ED, SWD) are reported there, not calibration metrics. Could the authors report SSR or empirical coverage on the stochastic benchmarks, or revise the claim to "better distributional fidelity"?

6. **Clarifications on architecture and formalism.**

    (a) Do `NF` and `NO` share weights or merely the same FNO design? They take different inputs and produce different output types, so weight sharing can be non-trivial.

    (b) What does "discrete encoders" mean in Section 4.1 --- tokenization into discrete tokens, or discrete diffusion over categorical latents?

    (c) Is $\mathcal{G}^\ddagger$ a random operator (interpretation a: equality in law of operator-valued random variables) or a deterministic map $\mathcal{A}\to\mathcal{P}(\mathcal{U})$ (interpretation b, consistent with how Section 4 implements it)?

    (d) The authors state that diffusion underfitting corresponds to epistemic uncertainty; could you provide further explanation of this?

**Limitations:**

Yes, the authors discuss limitations, including the restriction to regular-grid benchmarks (noted in the baselines section) and directions for future work (conformal calibration, inverse problems, irregular geometries). The Impact Statement is present but generic. No ethical concerns arise from this work.

**Strengths And Weaknesses:**

**Strengths**

1. **Clean, practical design.**
DLL is backbone-agnostic and inherits discretization invariance from the underlying NO. Fixing $r=64$ decouples inference cost from grid resolution.

2. **Meaningful theoretical grounding.**
Proposition 1 gives a $\mathcal{W}_2 \lesssim C\sqrt{\mathcal{L}_V}$ bound motivating velocity-matching; Proposition 2 identifies the operator encoder with the truncated KL subspace of the conditional covariance.

3. **Competitive empirical results.**
DLL achieves the best ED on both stochastic benchmarks and best NRMSE/CRPS on KS, with the probabilistic head appearing to also regularize deterministic rollout.

4. **Distinct contribution relative to concurrent work.**
The closest concurrent work, DINOZAUR (Matveev et al.), is FNO-specific and restricted to Gaussian posteriors; DLL is backbone-agnostic and supports non-Gaussian distributions.

---

**Weaknesses**

**A. Framing and claims.**

1. **Overclaimed plug-and-play framing.**
The abstract and introduction imply DLL can be attached to an already-trained NO. This is not the case: Stage 1 retrains $\mathtt{NO}\_\psi$ from scratch under $\mathcal{L}\_{\mathrm{OE}}$, changing both the output space and the objective, and Stage 2 is coupled to Stage 1 since the diffusion head operates on KL coefficients specific to $\mathtt{NO}\_\psi$. The framing should be corrected.

2. **Epistemic uncertainty claim is unsubstantiated and internally inconsistent.**
Section 3.3 claims DLL captures epistemic uncertainty via diffusion underfitting, yet Section 5.1 frames the stochastic benchmarks as evaluating aleatoric uncertainty with no mechanism to separate the two. Since $p_\theta(u\mid a)$ conflates both sources, the reported metrics (ED, SWD, $\mathrm{NRMSE}_s$) measure total predictive distribution quality, not aleatoric uncertainty alone. No experiment validates the epistemic claim (e.g., increasing uncertainty on out-of-distribution inputs). Additionally, the introduction asserts "better-calibrated uncertainty" on stochastic benchmarks, but ED, SWD, and $\mathrm{NRMSE}_s$ are distributional fidelity metrics, not calibration metrics; SSR appears only on deterministic benchmarks.

**B. Methodological concerns.**

1. **Inconsistency between input-independent coefficients and conditional flow model.**
Proposition 2 identifies the optimal coefficients as $\xi^\star_k(u,a) = \langle u, e_k(a)\rangle$, which depend on $a$ through the input-adapted eigenfunctions. However, $\mathtt{NF}_\varphi(u)$ has no access to $a$, so the training coefficients $\xi^{(i)} = \mathtt{NF}(u^{(i)})$ are produced without explicit reference to $a^{(i)}$. Since the basis $\mathtt{NO}(a)$ changes with $a$, the flow model learns $p(\xi \mid a)$ over coefficients that are not specific to the $a$-dependent basis, so there is no guarantee that samples $\xi \sim p(\xi \mid a)$ combined with $\mathtt{NO}(a)$ faithfully represent $p(u \mid a)$.

**C. Missing ablations and unexplained results.**

1. **Reconstruction comparison is unfair.**
Tables 6 and 7 compare OE and AE at mismatched compression ratios ($\times 2$ vs. $\times 4$ in 1D; $\times 16$ vs. $\times 256$ in 2D), always with OE at higher compression. The reported $20\times$ NRMSE gap on Burgers conflates encoder design with compression level; a matched-ratio comparison is needed to isolate the two effects.

2. **No ablation on the latent dimension $r$.**
The paper fixes $r=64$ across all benchmarks without ablating this choice. Given that the operator encoder is the information bottleneck, understanding how performance scales with $r$ is critical for assessing DLL's practical utility.

3. **Unexplained underperformance on Kolmogorov flow.**
On Kolmogorov flow (Table 5), pixel-space `DM` outperforms DLL on both NRMSE (0.369 vs. 0.426) and CRPS (0.692 vs. 0.822) with no explanation offered.

**D. Related work and positioning.**

1. **Incomplete related work and missing baselines.**
Phillips et al. use a score-based model in the KL spectral coefficient space---the same conceptual basis as DLL---but are not discussed. Among supervised contemporaries, DINOZAUR (Matveev et al.), CoNFiLD (Du et al.), and WDNO (Hu et al.) are closely related but absent from the comparisons. The Neural Process family---including Gridded TNPs (Ashman et al.), Neural Diffusion Processes (Dutordoir et al.), and the Gaussian Neural Process (Bruinsma et al.)---is directly relevant to probabilistic operator learning and entirely missing. Millard et al. on conformal prediction for neural operators warrants at least a citation.

**E. Minor presentation issues.**

1. **Notation inconsistency in Table 1.** The `DM` row reads $\mathcal{G}_\theta(a) \stackrel{d}{\approx} \mathcal{G}^\dagger(a)$ (deterministic dagger), while `LDM` and `DLL` read $\stackrel{d}{\approx} \mathcal{G}^\ddagger(a)$ (stochastic double-dagger). Since pixel-space diffusion is also evaluated on stochastic benchmarks, this appears to be a typographical error.

2. **Missing U-Net architecture details for DM and LDM.** The appendix provides no architectural specifics for the U-Net backbones used in the `DM` and `LDM` baselines (number of layers, channel widths, kernel sizes, skip connection design, etc.) which can significantly affect the U-Net performance, and are thus important for reproducibility.

3. **Overstated limitations of Bayesian surrogates (Sections 3.2--3.3).** Sections 3.2--3.3 conflate limitations of specific Bayesian approximations with fundamental limitations of Bayesian inference: (i) the non-Gaussian limitation applies to Gaussian variational families, not Bayesian inference in general; (ii) input-conditioned and hierarchical priors are standard, so posteriors are not generically "independent of the input"; and (iii) the failure to model aleatoric uncertainty reflects a choice of likelihood family, not a structural property of Bayesianism.

---

**References**

1. Matveev, A. et al. (2025). Light-Weight Diffusion Multiplier and Uncertainty Quantification for Fourier Neural Operators. https://arxiv.org/abs/2508.00643

2. Phillips, A. et al. (2022). Spectral Diffusion Processes. *Score-based Methods Workshop, NeurIPS 2022*. https://arxiv.org/abs/2209.14125

3. Du, P. et al. (2024). CoNFiLD: Conditional Neural Field Latent Diffusion Model Generating Spatiotemporal Turbulence. https://arxiv.org/abs/2403.05940

4. Hu, P. et al. (2025). Wavelet Diffusion Neural Operator. *ICLR 2025*. https://arxiv.org/abs/2412.04833

5. Garnelo, M. et al. (2018). Conditional Neural Processes. https://arxiv.org/abs/1807.01613

6. Garnelo, M. et al. (2018). Neural Processes. https://arxiv.org/abs/1807.01622

7. Bruinsma, W.P. et al. (2021). The Gaussian Neural Process. *3rd Symposium on Advances in Approximate Bayesian Inference*. https://arxiv.org/abs/2101.03606

8. Ashman, M. et al. (2024). Gridded Transformer Neural Processes for Large Unstructured Spatio-Temporal Data. https://arxiv.org/abs/2410.06731

9. Ma, Z. et al. (2024). Calibrated Uncertainty Quantification for Operator Learning via Conformal Prediction. https://arxiv.org/abs/2402.01960

10. Dutordoir, V. et al. (2023). Neural Diffusion Processes. *ICML 2023*.

11. Millard, D. et al. (2025). Split Conformal Prediction in the Function Space with Neural Operators. https://arxiv.org/abs/2509.04623

---

> ### Author Rebuttal · Authors · 2026-03-30
>
> We appreciate the reviewer's valuable comments.
>
> **Q1: Sliced $\mathcal{W}_2$**
>
> We additionally report sliced $\mathcal{W}_2$ (to be added to the paper). For a minor note, since $\mathcal{W}_1 \le \mathcal{W}_2$, our theory also bounds $\mathcal{W}_1$.
>
> | Models | Burgers | Darcy |
> | - | - | - |
> | FNO | 0.815 | 0.022 |
> | FNO-d | 0.562 | 0.020 |
> | PNO | 0.444 | 0.012 |
> | DM | 0.376 | 0.011 |
> | LDM | 0.389 | 0.010 |
> | DLL | 0.361 | 0.009 |
>
> **Q2 and W.C.1: Reconstruction Bottleneck**
>
> The compression mismatch stems from architectural differences. U-Net AEs use spatial downsampling, yielding resolution-independent compression, whereas OE outputs a fixed-size vector, so compression scales with resolution. Additionally, AE is unconditional, while OE is input-conditioned, which is essential for operator learning, making AE not directly compatible with DLL.
>
> While smaller compression improves reconstruction, it does not necessarily improve distributional fidelity or diffusion performance (as seen in latent diffusion and our $r$ ablations). Thus, the reconstruction gap does not necessarily indicate a bottleneck.
>
> **Q3 and W.C.2: Ablations on varying $r$**
>
> We conduct ablations with varying $r$ and we refer to the response to the **Reviewer rSeH, Q1** for details.
>
> **Q4 and W.C.3: Underperformance on Kolmogorov flow**
>
> We note that our goal is not to outperform all generative baselines, but to provide a modular method for incorporating uncertainty into neural operators.
>
> The underperformance on Kolmogorov flow can be attributed to architectural differences. Pixel-space DM uses a U-Net with strong inductive biases for capturing fine-scale spatial structures, while DLL performs diffusion in a low-dimensional coefficient space, which may limit expressivity for highly complex dynamics.
>
>
> **Q5 and W.A.2: Calibration on stochastic benchmarks**
>
> We thank the reviewer for the clarification. We will revise the claim to "better distributional fidelity". In the stochastic benchmarks, our first goal is to evaluate whether DLL captures the conditional output distribution induced by the stochastic forcing, and we therefore focus on distributional metrics.
>
> Achieving calibration of full predictive uncertainty in function space, which involves both aleatoric and epistemic components, is a broader objective for diffusion based generative models and left for future work.
>
> **Q6: Clarifications on architecture and formalism**
>
> (a) NF and NO do not share weights; they only use the same FNO architecture for simplicity. They take different inputs and serve different roles.
>
> (b) By "discrete encoders", we meant encoders defined on a fixed discretization of the function space. We agree the term may be misleading and will revise it.
>
> (c) As mentioned in Section 2.1, our intent is to treat $\mathcal{G}^\ddagger$ as a deterministic map from $\mathcal{A}$ to probability measures on the function space $\mathcal{U}$. We will revise Section 2.1 to make this explicit.
>
> (d) (also related with **W.A.2**) Our claim is heuristic and primarily supported in the deterministic setting, where uncertainty is mainly epistemic and our experiments suggest that underfitted residual diffusion spread can be a useful proxy. In stochastic benchmarks, the same mechanism may also contribute to uncertainty estimates, but there the predictive distribution mixes aleatoric and epistemic effects, so we do not claim a clean separation or perfect calibration. We will revise the text to make this scope explicit.
>
> **W.A.1: Plug-and-play framing**
>
> DLL supports arbitrary NO backbones but is not post-hoc. We will clarify that “plug-and-play” is architectural only.
>
> **W.B: Inconsistency coefficient and flow model**
>
> We agree that the optimal coefficients $\xi_k^\star(u,a)=\langle u, e_k(a)\rangle$ depend on $a$, and we originally considered a joint encoder $\xi=\mathtt{NF}(u,a)$. In practice, however, we found that augmenting the encoder with $a$ did not yield measurable empirical gains, while increasing complexity. We therefore use $\xi=\mathtt{NF}(u)$ with an $a$-dependent decoder $\Phi(a)$ as a practical simplification, and we will clarify that this is an empirical design choice rather than the theoretically most general formulation.
>
>
> **W.D: Missing references**
>
> We appreciate the reviewer and will include these works. Phillips et al. are related via spectral generative modeling but do not address conditional operator learning. DINOZAUR focuses on Bayesian UQ, while CoNFiLD and WDNO target spatiotemporal forecasting. Neural Processes are also related but are typically framed as regression on observation sets rather than operator learning.
>
> **W.E: Presentation issues**
>
> 1. The notation in Table 1 contains a typographical error, and we will correct.
>
> 2. We will include full architectural details of the U-Net backbones.
>
> 3. We will revise these sections to clarify.

---

> > ### Author Rebuttal · Reviewer_dXc6 · 2026-04-03
> >
> > I thank the authors for their response and for their willingness to revise the framing around calibration, epistemic uncertainty, and the plug-and-play characterisation. I maintain my score of **4**. The core contribution---and in particular its connection to the Karhunen--Lo\`eve expansion and classical functional data analysis---remains a compelling and self-contained contribution.
> >
> > I do, however, think the submission could be further strengthened on two points.
> >
> > **1. Baseline coverage.**
> > I appreciate the authors' engagement with the missing references and recognise that exhaustive baseline comparison is not always feasible. That said, I gently push back on some of the exclusion rationale.
> >
> > - *CoNFiLD and WDNO.* The central problem---learning a conditional generative model mapping an input function to a distribution over output functions---is shared with the present work; the distinction lies primarily in what is treated as input versus output. I believe both methods are applicable to the benchmarks considered here.
> > - *Neural processes.* I agree the framing differs, but the underlying task of function-to-function regression aligns closely with this paper's setting. Operator learning has moreover been explored within the NP framework---notably in the Convolutional Conditional Neural Process (Gordon et al., 2020) and the Spectral Convolutional Conditional Neural Process (Mohseni & Duffield, 2024). I also note an internal tension in the exclusion logic: if NPs are set aside on the grounds that they do not specifically target operator learning, the same argument would apply to the included DM and LDM baselines.
> >
> > Including even one or two of these methods would meaningfully help readers situate DLL's contribution within the broader landscape.
> >
> > **2. Hypothesis for DLL underperformance.**
> > To be clear, I do not view underperformance on some benchmarks as a weakness that undermines the paper, and I did not flag it as such in my review. The paper offers genuine new insights regardless. That said, because the authors have the deepest familiarity with DLL's behaviour, even a brief hypothesis as to what might drive the observed underperformance would be of considerable value to future readers.
> >
> > **References**
> >
> > Gordon, J. et al. (2020). Convolutional Conditional Neural Processes. *ICLR 2020*. https://arxiv.org/abs/1910.13556
> >
> > Mohseni & Duffield. (2024). Spectral Convolutional Conditional Neural Processes. https://arxiv.org/abs/2404.13182

---

> > > ### Author Response · Authors · 2026-04-06
> > >
> > > We sincerely thank the reviewer for the thoughtful follow-up and the constructive suggestions.
> > >
> > > **Baseline coverage**
> > >
> > > We fully agree that CoNFiLD, WDNO, and neural processes are relevant to our problem setting and are applicable in principle, and that broader baseline coverage would provide additional perspective. Our baseline selection, however, was designed to isolate two specific comparison axes: (i) whether attaching a probabilistic head improves over deterministic and lightweight probabilistic variants built on the same neural-operator backbone, and (ii) whether diffusion in a low-dimensional coefficient space can remain competitive with more standard diffusion settings in pixel space or autoencoder latent space. For this reason, we chose FNO/FNO-d/PNO as backbone-matched baselines to isolate the effect of the probabilistic head, and DM/LDM as diffusion-space baselines.
> > >
> > > At the same time, applying these methods in our setting would require some modification relative to their original formulations. WDNO is developed for diffusion in the wavelet domain over full spatiotemporal fields, whereas our benchmarks include next-step or terminal-time prediction and a steady elliptic Darcy problem with no temporal axis. CoNFiLD focuses on generative modeling of full fields, with conditioning handled at sampling time rather than through an amortized conditional operator map learned from paired input-output data. Neural processes are also highly relevant, but the cited ConvCNP/SConvCNP formulations are built around context-query regression from observations of the target function itself, so applying them in our setting would require reformulating them for input-function-conditioned operator learning. For these reasons, and under our limited training resources, we prioritized baselines that more directly matched the experimental questions studied in this work.
> > >
> > > **Hypothesis for the underperformance on Kolmogorov flow**
> > >
> > > We also appreciate the reviewer's suggestion to provide some intuition for the underperformance on Kolmogorov flow. Our current hypothesis is that DLL primarily improves the rollout stability of the underlying neural operator backbone, while its overall performance ceiling still depends strongly on the quality of that backbone. This is consistent with DLL's role as a lightweight probabilistic head attached to a backbone NO, and with our broader empirical observation that DLL improves the rollout stability of the corresponding deterministic backbone. On KS, where the FNO backbone is already competitive, DLL yields a further gain. On Kolmogorov flow, however, the FNO backbone itself trails the pixel-space DM by a substantial margin, so although DLL improves over FNO, it does not fully close the gap to DM.

---

### Decision · Program_Chairs · 2026-04-30

**Decision:**

Accept (regular)

**Comment:**

The paper introduces the Diffusion Last Layer (DLL), a flexible probabilistic head for neural operators.
All four reviewers recommend accepting this paper, recognizing its technically solid contribution to uncertainty quantification in scientific computing.
Reviewers praised the well-motivated methodology, elegantly leveraging a low-rank Karhunen-Loève expansion in function space to enable efficient, discretization-invariant conditional generation.
The empirical results demonstrate strong performance across both stochastic and deterministic PDE benchmarks, and the method offers a favorable trade-off between accuracy, distributional fidelity and stability while remaining highly modular.

During the review process, the reviewers raised questions regarding empirical ablations, architectural generalizability, framing, and the practical utility of generative models for forward problems.
The authors provided a rebuttal that addressed these concerns, by supplying new ablations on dataset size, latent rank, and diffusion steps, and demonstrating DLL's compatibility with a DeepONet backbone on an inverse problem.
They also agreed to temper their claims regarding "plug-and-play" modularity and "epistemic uncertainty," appropriately reframing the latter around "distributional fidelity."
While reviewers noted that additional baselines (like Neural Processes) or a stronger focus on inverse problems could further elevate the work, the committee ultimately agreed that the authors' clarifications, theoretical grounding, and efficiency arguments sufficiently validated the core methodological contribution.

For the final version, the authors should incorporate the additions and revisions promised during the rebuttal. This includes adding the ablation studies on dataset size, latent rank, and diffusion steps, as well as the DeepONet compatibility experiments on the Darcy flow inverse problem.
The text should be updated to correct the overclaimed "plug-and-play" language, clarify the scope of epistemic uncertainty, and revise "calibration" claims to "distributional fidelity."
Additionally, the authors should add the discussed related works (including Neural Processes, CoNFiLD, WDNO, and others mentioned by the reviewers) and include their provided hypotheses regarding the underperformance on Kolmogorov flow and the broader trade-offs of using generative methods for forward versus inverse problems.